# No MCMC for me: Amortized sampling for fast and stable training of energy-based models

**Will Grathwohl**[*]
University of Toronto & Vector Institute
Google Research
wgrathwohl@cs.toronto.edu

**Jacob Kelly**[*]
University of Toronto & Vector Institute
jkelly@cs.toronto.edu

**Milad Hashemi**
Google Research
miladh@google.com

**Mohammad Norouzi & Kevin Swersky**
Google Research
{mnorouzi, kswersky}@google.com

**David Duvenaud**
University of Toronto & Vector Institute
duvenaud@cs.toronto.edu

## Abstract

Energy-Based Models (EBMs) present a flexible and appealing way to represent uncertainty. Despite recent advances, training EBMs on high-dimensional data remains a challenging problem as the state-of-the-art approaches are costly, unstable, and require considerable tuning and domain expertise to apply successfully. In this work we present a simple method for training EBMs at scale which uses an entropy-regularized generator to amortize the MCMC sampling typically used in EBM training. We improve upon prior MCMC-based entropy regularization methods with a fast variational approximation. We demonstrate the effectiveness of our approach by using it to train tractable likelihood models. Next, we apply our estimator to the recently proposed Joint Energy Model (JEM), where we match the original performance with faster and stable training. This allows us to extend JEM models to semi-supervised classification on tabular data from a variety of continuous domains.

## 1 Introduction

Energy-Based Models (EBMs) have recently regained popularity within machine learning, partly inspired by the impressive results of Du & Mordatch (2019) and Song & Ermon (2020) on large-scale image generation. Beyond image generation, EBMs have also been successfully applied to a wide variety of applications including: out-of-distribution detection (Grathwohl et al., 2019; Du & Mordatch, 2019; Song & Ou, 2018), adversarial robustness (Grathwohl et al., 2019; Hill et al., 2020; Du & Mordatch, 2019), reliable classification (Grathwohl et al., 2019; Liu & Abbeel, 2020) and semi-supervised learning (Song & Ou, 2018; Zhao et al.). Strikingly, these EBM approaches outperform alternative classes of generative models and rival hand-tailored solutions on each task.

Despite progress, training EBMs is still a challenging task. As shown in Table 1, existing training methods are all deficient in at least one important practical aspect. Markov chain Monte Carlo (MCMC) methods are slow and unstable during training (Nijkamp et al., 2019a; Grathwohl et al., 2020). Score matching mechanisms, which minimize alternative divergences are also unstable and most methods cannot work with discontinuous nonlinearities (such as ReLU) (Song & Ermon, 2019b; Hyvärinen, 2005; Song et al., 2020; Pang et al., 2020b; Grathwohl et al., 2020; Vincent, 2011). Noise contrastive approaches, which learn energy functions through density ratio estimation, typically don't scale well to high-dimensional data (Gao et al., 2020; Rhodes et al., 2020; Gutmann & Hyvärinen, 2010; Ceylan & Gutmann, 2018).

---

[*]Equal Contribution. Code available at github.com/wgrathwohl/VERA

| Training Method | Fast | Stable training | High dimensions | No aux. model | Unrestricted architecture | Approximates likelihood |
|---|---|---|---|---|---|---|
| Markov chain Monte Carlo | ✗ | ✗ | ✓ | ✓ | ✓ | ✓ |
| Score Matching Approaches | ✓ | ✗ | ✓ | ✓ | ✗ | ✗ |
| Noise Contrastive Approaches | ✓ | ✓ | ✗ | ✗ | ✓ | ✗ |
| VERA (ours) | ✓ | ✓ | ✓ | ✗ | ✓ | ✓ |

Table 1: Features of EBM training approaches. Trade-offs must be made when training unnormalized models and no approach to date satisfies all of these properties.

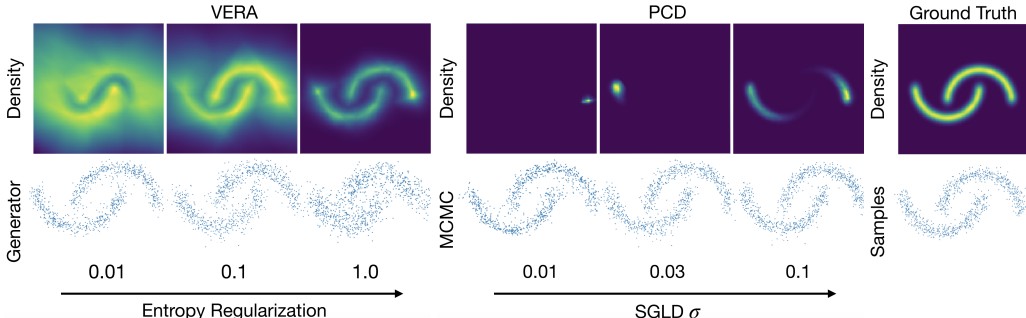

Figure 1: Comparison of EBMs trained with VERA and PCD. We see that as entropy regularization goes to 1, the density becomes more accurate. For PCD, all samplers produce high quality samples, but low-quality density models as the distribution of MCMC samples may be arbitrarily far away from the model density.

In this work, we present a simple method for training EBMs which performs as well as previous methods while being faster and substantially easier to tune. Our method is based on reinterpreting maximum likelihood as a bi-level variational optimization problem, which has been explored in the past for EBM training (Dai et al., 2019). This perspective allows us to amortize away MCMC sampling into a GAN-style generator which is encouraged to have high entropy. We accomplish this with a novel approach to entropy regularization based on a fast variational approximation. This leads to the method we call Variational Entropy Regularized Approximate maximum likelihood (VERA).

Concretely, we make the following contributions:

- We improve the MCMC-based entropy regularizer of Dieng et al. (2019) with a parallelizable variational approximation.
- We show that an entropy-regularized generator can be used to produce a variational bound on the EBM likelihood which can be optimized more easily than MCMC-based estimators.
- We demonstrate that models trained in this way achieve much higher likelihoods than methods trained with alternative EBM training procedures.
- We show that our approach stabilizes and accelerates the training of recently proposed Joint Energy Models (Grathwohl et al., 2019).
- We show that the stabilization of our approach allows us to use JEM for semi-supervised learning, outperforming virtual adversarial training when little prior domain knowledge is available (e.g., for tabular data).

## 2 ENERGY BASED MODELS

An energy-based model (EBM) is any model which parameterizes a density as

$$p_\theta(x) = \frac{e^{f_\theta(x)}}{Z(\theta)} \tag{1}$$

where $f_\theta : \mathbb{R}^D \to \mathbb{R}$ and $Z(\theta) = \int e^{f_\theta(x)} \mathrm{d}x$ is the normalizing constant which is not explicitly modeled. Any probability distribution can be represented in this way for some $f_\theta$. The energy-based

parameterization has been used widely for its flexibility, ease of incorporating known structure, and relationship to physical systems common in chemistry, biology, and physics (Ingraham et al., 2019; Du et al., 2020; Noé et al., 2019).

The above properties make EBMs an appealing model class, but because they are unnormalized many tasks which are simple for alternative model classes become challenging for EBMs. For example, exact samples cannot be drawn and likelihoods cannot be exactly computed (or even lower-bounded). This makes training EBMs challenging as we cannot simply train them to maximize likelihood. The most popular approach to train EBMs is to approximate the gradient of the maximum likelihood objective. This gradient can be written as:

$$\nabla_\theta \log p_\theta(x) = \nabla_\theta f_\theta(x) - \mathbb{E}_{p_\theta(x')}[\nabla_\theta f_\theta(x')]. \tag{2}$$

MCMC techniques are used to approximately generate samples from $p_\theta(x)$ (Tieleman, 2008). Practically, this approach suffers from poor stability and computational challenges from sequential sampling. Many tricks have been developed to overcome these issues (Du & Mordatch, 2019), but they largely still persist. Alternative estimators have been proposed to circumvent these challenges, including score matching (Hyvärinen, 2005), noise contrastive estimation (Gutmann & Hyvärinen, 2010), and variants thereof. These suffer from their own challenges in scaling to high dimensional data, and sacrifice the statistical efficiency of maximum likelihood.

In Figure 1 we visualize densities learned with our approach and Persistent Contrastive Divergence (Tieleman, 2008) (PCD) training. As we see, the sample quality of the PCD models is quite high but the learned density models do not match the true model. This is due to accrued bias in the gradient estimator from approximate MCMC sampling (Grathwohl et al., 2020). Prior work (Nijkamp et al., 2019b) has argued that this objective actually encourages the approximate MCMC samples to match the data rather than the density model. Conversely, we see that our approach (with proper entropy regularization) recovers a high quality model.

## 3 Variational Maximum Likelihood

We seek the energy function which maximizes likelihood given in Equation 1. We examine the intractable component of the log-likelihood, the log partition-function $\log Z(\theta) = \log \int e^{f_\theta(x)} \mathrm{d}x$. We can re-write this quantity as the optimum of

$$\log Z(\theta) = \max_q \mathbb{E}_{q(x)}[f_\theta(x)] + H(q) \tag{3}$$

where $q$ is a distribution and $H(q) = -\mathbb{E}_{q(x)}[\log q(x)]$ denotes its entropy [1] (see the Appendix A.1 for the derivation). Plugging this into our original maximum likelihood statement we obtain:

$$\hat{\theta} = \operatorname*{argmax}_\theta \left[ \mathbb{E}_{p_{\text{data}}(x)}[f_\theta(x)] - \max_q \left[ \mathbb{E}_{q(x)}[f_\theta(x)] + H(q) \right] \right] \tag{4}$$

which gives us an alternative method for training EBMs. We introduce an auxiliary sampler $q_\phi$ which we train online to optimize the inner-loop of Equation 4. This objective was used for EBM training in Kumar et al. (2019); Abbasnejad et al. (2019); Dai et al. (2017), Dai et al. (2019) (motivated by Fenchel Duality (Wainwright & Jordan, 2008)). Abbasnejad et al. (2019) use an implicit generative model and Dai et al. (2019) propose to use a sampler which is inspired by MCMC sampling from $p_\theta(x)$ and whose entropy can be computed exactly.

Below we describe our approach which utilizes the same objective with a simpler sampler and a new approach to encourage high entropy. We note that when training $p_\theta(x)$ and $q(x)$ online together, the inner maximization will not be fully optimized. This leads our training objective for $p_\theta(x)$ to be an *upper* bound on $\log p_\theta(x)$. In Section 5.1 we explore the impact of this fact and find that the bound is tight enough to train models that achieve high likelihood on high-dimensional data.

---

[1]For continuous spaces, this would be the differential entropy, but we simply use entropy here for brevity.

## 4 METHOD

We now present a method for training an EBM $p_\theta(x) = e^{f_\theta(x)}/Z(\theta)$ to optimize Equation 4. We introduce a generator distribution of the form $q_\phi(x) = \int_z q_\phi(x|z)q(z)\mathrm{d}z$ such that:

$$q_\phi(x|z) = \mathcal{N}(g_\psi(z), \sigma^2 I), \qquad q(z) = \mathcal{N}(0, I) \tag{5}$$

where $g_\psi$ is a neural network with parameters $\psi$ and thus, $\phi = \{\psi, \sigma^2\}$. This is similar to the decoder of a variational autoencoder (Kingma & Welling, 2013). With this architecture it is easy to optimize the first and second terms of Equation 4 with reparameterization, but the entropy term requires more care.

### 4.1 ENTROPY REGULARIZATION

Estimating entropy or its gradients is a challenging task. Multiple, distinct approaches have been proposed in recent years based on Mutual Information estimation (Kumar et al., 2019), variational upper bounds (Ranganath et al., 2016), Denoising Autoencoders (Lim et al., 2020), and nearest neighbors (Singh & Póczos, 2016).

The above methods require the training of additional auxiliary models or do not scale well to high dimensions. Most relevant to this work are Dieng et al. (2019); Titsias & Ruiz (2019) which present a method for encouraging generators such as ours to have high entropy by estimating $\nabla_\phi H(q_\phi)$. The estimator takes the following form:

$$
\begin{aligned}
\nabla_\phi H(q_\phi) &= \nabla_\phi \mathbb{E}_{q_\phi(x)}[\log q_\phi(x)] \\
&= \nabla_\phi \mathbb{E}_{p(z)p(\epsilon)}[\log q_\phi(x(z,\epsilon))] \qquad \text{(Reparameterize sampling)} \\
&= \mathbb{E}_{p(z)p(\epsilon)}[\nabla_\phi \log q_\phi(x(z,\epsilon))] \\
&= \mathbb{E}_{p(z)p(\epsilon)}[\nabla_x \log q_\phi(x(z,\epsilon))^T \nabla_\phi x(z,\epsilon)] \qquad \text{(Chain rule)}
\end{aligned} \tag{6}
$$

where we have written $x(z,\epsilon) = g_\psi(z) + \sigma\epsilon$. All quantities in Equation 6 can be easily computed except for the score-function $\nabla_x \log q_\phi(x)$. The following estimator for this quantity can be easily derived (see Appendix A.2):

$$\nabla_x \log q_\phi(x) = \mathbb{E}_{q_\phi(z|x)}[\nabla_x \log q_\phi(x|z)] \tag{7}$$

which requires samples from the posterior $q_\phi(z|x)$ to estimate. Dieng et al. (2019); Titsias & Ruiz (2019) generate these samples using Hamiltonian Monte Carlo (HMC) (Neal et al., 2011), a gradient-based MCMC algorithm. As used in Dieng et al. (2019), 28 *sequential* gradient computations must be made per training iteration. Since a key motivation of this work is to circumvent the costly sequential computation of MCMC sampling, this is not a favourable solution. In our work we propose a more efficient solution that we find works just as well empirically.

### 4.2 VARIATIONAL APPROXIMATION WITH IMPORTANCE SAMPLING

We propose to replace HMC sampling of $q_\phi(z|x)$ with a variational approximation $\xi(z \mid z_0) \approx q_\phi(z \mid x)$ where $z_0$ is a conditioning variable we will define shortly. We can use this approximation with self-normalized importance sampling to estimate

$$
\begin{aligned}
\nabla_x \log q_\phi(x) &= \mathbb{E}_{q_\phi(z|x)}[\nabla_x \log q_\phi(x \mid z)] \\
&= \mathbb{E}_{p(z_0)q_\phi(z|x)}[\nabla_x \log q_\phi(x \mid z)] \\
&= \mathbb{E}_{p(z_0)\xi(z|z_0)}\left[\frac{q_\phi(z \mid x)}{\xi(z \mid z_0)}\nabla_x \log q_\phi(x \mid z)\right] \\
&= \mathbb{E}_{p(z_0)\xi(z|z_0)}\left[\frac{q_\phi(z, x)}{q_\phi(x)\xi(z \mid z_0)}\nabla_x \log q_\phi(x \mid z)\right] \\
&\approx \sum_{i=1}^{k} \frac{w_i}{\sum_{j=1}^{k} w_j}\nabla_x \log q_\phi(x \mid z_i) \equiv \widetilde{\nabla}_x \log q_\phi(x; \{z_i\}_{i=1}^{k}, z_0)
\end{aligned} \tag{8}
$$

where $\{z_i\}_{i=1}^{k} \sim \xi(z \mid z_0)$ and $w_i \equiv \frac{q_\phi(z_i, x)}{\xi(z_i|z_0)}$. We use $k = 20$ importance samples for all experiments in this work. This approximation holds for any conditioning information we would like to use.

To choose this, let us consider how samples $x \sim q_\phi(x)$ are drawn. We first sample $z_0 \sim \mathcal{N}(0, I)$ and then $x \sim q_\phi(x \mid z_0)$. In our estimator we want a variational approximation to $q_\phi(z \mid x)$ and by construction, $z_0$ is a sample from this distribution. For this reason we let our variational approximation be

$$\xi_\eta(z \mid z_0) = \mathcal{N}(z \mid z_0, \eta^2 I), \tag{9}$$

or simply a diagonal Gaussian centered at the $z_0$ which generated $x$. For this approximation to be useful we must tune the variance $\eta^2$. We do this by optimizing the standard Evidence Lower-Bound at every training iteration

$$\mathcal{L}_{\text{ELBO}}(\eta; z_0, x) = \mathbb{E}_{\xi_\eta(z|z_0)} \left[ \log(q_\phi(x \mid z)) + \log(q_\phi(z)) \right] + H(\xi_\eta(z \mid z_0)). \tag{10}$$

We then use $\xi_\eta(z \mid z_0)$ to approximate $\nabla_x \log q_\phi(x)$ which we plug into Equation 6 to estimate $\nabla_\phi H(q_\phi)$ for training our generator. A full derivation and discussion can be found in Appendix A.3.

Combining the tools presented above we arrive at our proposed method which we call Variational Entropy Regularized Approximate maximum likelihood (VERA), outlined in Algorithm 1. We found it helpful to further add an $\ell_2$-regularizer with weight $0.1$ to the gradient of our model's likelihood as in Kumar et al. (2019). In some of our larger-scale experiments we reduced the weight of the entropy regularizer as in Dieng et al. (2019). We refer to the entropy regularizer weight as $\lambda$.

---

**Algorithm 1:** VERA Training

---

**Input** : EBM $p_\theta(x) \propto e^{f_\theta(x)}$, generator $q_\phi(x, z)$, approximate posterior $\xi_\eta(z|z_0)$,
           entropy weight $\lambda$, gradient penalty $\gamma = .1$
**Output:** Parameters $\theta$ such that $p_\theta \approx p$
**while** *True* **do**
    Sample mini-batch $x$, and generate mini-batch $x_g, z_0 \sim q_\phi(x, z)$
    Compute $\mathcal{L}_{\text{ELBO}}(\eta; z_0, x_g)$ and update $\eta$             // Update posterior
    Compute $\log f_\theta(x) - \log f_\theta(x_g) + \gamma ||\nabla_x \log f_\theta(x)||^2$ and update $\theta$    // Update EBM
    Sample $\{z_i\}_{i=1}^k \sim \xi_\eta(z|z_0)$
    Compute $s = \widetilde{\nabla}_x \log q_\phi(x; \{z_i\}_{i=1}^k, z_0)$       // Estimate score fn (Eq.8)
    Compute $g = s^T \nabla_\phi x_g$       // Estimate entropy gradient (Eq.6)
    Update $\phi$ with $\nabla_\phi \log f_\theta(x_g) + \lambda g$         // Update generator
**end**

---

## 5 EBM Training Experiments

We present results training various models with VERA and related approaches. In Figure 1 we visualize the impact of our generator's entropy on the learned density model and compare this with MCMC sampling used in PCD learning. In Section 5.1, we explore this quantitatively by training tractable models and evaluating with test-likelihood. In Section 5.2 we explore the bias of our entropy gradient estimator and the estimator's effect on capturing modes.

### 5.1 Fitting Tractable Models

Optimizing the generator in VERA training minimizes a variational *upper* bound on the likelihood of data under our model. If this bound is not sufficiently tight, then training the model to maximize this bound will not actually improve likelihood. To demonstrate the VERA bound is tight enough to train large-scale models we train NICE models (Dinh et al., 2014) on the MNIST dataset. NICE is a normalizing flow (Rezende & Mohamed, 2015) model – a flexible density estimator which enables both exact likelihood computation and exact sampling. We can train this model with VERA (which does not require either of these abilities), evaluate the learned model using likelihood, and generate exact samples from the trained models. Full experimental details[2] can be found in Appendix B.3 and additional results can be found in Appendix C.1.

---

[2]This experiment follows the NICE experiment in Song et al. (2020) and was based on their implementation.

We compare the performance of VERA with maximum likelihood training as well as a number of approaches for training unnormalized models; Maximum Entropy Generators (MEG) (Kumar et al., 2019), Persistent Contrastive Divergence (PCD), Sliced Score Matching (SSM) (Song et al., 2020), Denoising Score Matching (DSM) (Vincent, 2011), Curvature Propagation (CP-SM) (Martens et al., 2012), and CoopNets (Xie et al., 2018). As an ablation we also train VERA with the HMC-based entropy regularizer of Dieng et al. (2019), denoted VERA-(HMC). Table 2 shows that VERA outperforms all approaches that do not require a normalized model.

Figure 2 shows exact samples from our NICE models. For PCD we can see (as observed in Figure 1) that while the approximate MCMC samples resemble the data distribution, the true samples from the model do not. This is further reflected in the reported likelihood value which falls behind all methods besides DSM. CoopNets perform better than PCD, but exhibit the same behavior of generator samples resembling the data, but not matching true samples. We attribute this behavior to the method's reliance on MCMC sampling.

Conversely, models trained with VERA generate coherent and diverse samples which reasonably capture the data distribution. We also see that the samples from the learned generator

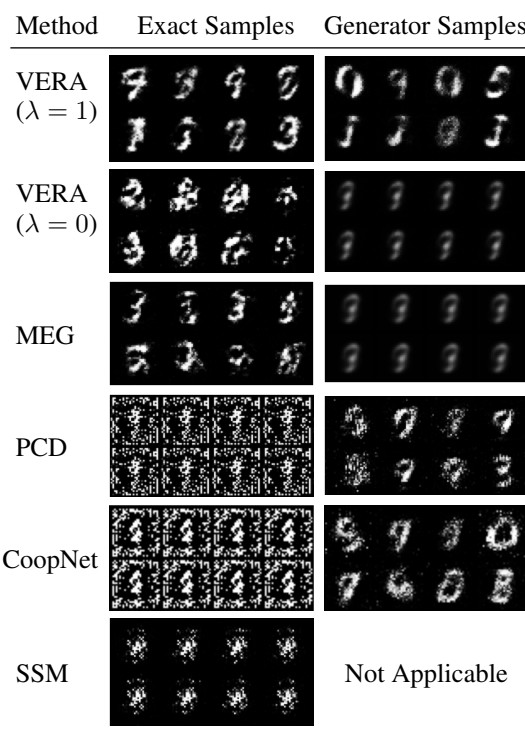

Figure 2: *Left:* Exact samples from NICE model trained with various methods. *Right:* Approximate samples used for training. For VERA, MEG, and CoopNet, these come from the generator, for PCD these are approximate MCMC samples.

much more closely match true samples from the NICE model than PCD and MEG. When we remove the entropy regularizer from the generator ($\lambda = 0.0$) we observe a considerable decrease in likelihood and we find that the generator samples are far less diverse and do not match exact samples at all. Intriguingly, entropy-free VERA outperforms most other methods. We believe this is because even without the entropy regularizer we are still optimizing a (weak) bound on likelihood. Conversely, the score-matching methods minimize an alternative divergence which will not necessarily correlate well with likelihood. Further, Figure 2 shows that MEG performs on par with entropy-free VERA indicating that the Mutual Information-based entropy estimator may not be accurate enough in high dimensions to encourage high entropy generators.

| Maximum Likelihood | VERA $\lambda = 1.0$ | $\lambda = 0.0$ | VERA (HMC) $\lambda = 1.0$ | MEG | PCD | SSM | DSM | CP-SM | CoopNet |
|---|---|---|---|---|---|---|---|---|---|
| -791 | **-1138** | -1214 | -1165 | -1219 | -4207 | -2039 | -4363 | -1517 | -1465 |

Table 2: Fitting NICE models using various learning approaches for unnormalized models. Results for SSM, DCM, CP-SM taken from Song et al. (2020).

## 5.2 UNDERSTANDING OUR ENTROPY REGULARIZER

In Figure 3, we explore the quality of our score function estimator on a PCA (Tipping & Bishop, 1999) model fit to MNIST, a setting where we can compute the score function exactly (see Appendix B.4 for details). The importance sampling estimator (with 20 importance samples) has somewhat larger variance than the HMC-based estimator but has a notably lower bias of .12. The HMC estimator using 2 burn-in steps (recommended in Dieng et al. (2019)) has a bias of .48. Increasing the burn-in steps to 500 reduces the bias to .20 while increasing variance. We find the additional variance of our estimator is remedied by mini-batch averaging and the reduced bias helps explain the improved performance in Table 2.

Further, we compute the effective sample size (Kong, 1992) (ESS) of our importance sampling proposal on our CIFAR10 and MNIST models and achieve an ESS of 1.32 and 1.29, respectively using 20 importance samples. When an uninformed proposal ($\mathcal{N}(0, I)$) is used, the ESS is 1.0 for both models. This indicates our gradient estimates are informative for training. More details can be found in Appendix B.6.

Next, we count the number of modes captured on a dataset with 1,000 modes consisting of 3 MNIST digits stacked on top of one another (Dieng et al., 2019; Kumar et al., 2019). We find that both VERA and VERA (HMC) recover 999 modes, but training with *no entropy regularization* recovers 998 modes. We conclude that entropy regularization is unnecessary for preventing mode collapse in this setting.

# 6  APPLICATIONS TO JOINT ENERGY MODELS

Joint Energy Models (JEM) (Grathwohl et al., 2019) are an exciting application of EBMs. They reinterpret standard classifiers as EBMs and train them as such to create powerful hybrid generative/discriminative models which improve upon purely-discriminative models at out-of-distribution detection, calibration, and adversarial robustness.

Traditionally, classification tasks are solved with a function $f_\theta : \mathbb{R}^D \to \mathbb{R}^K$ which maps from the data to $K$ unconstrained real-valued outputs (where $k$ is the number of classes). This function parameterizes a distribution over labels $y$ given data $x$: $p_\theta(y|x) = e^{f(x)[y]}/\sum_{y'} e^{f_\theta(x)[y']}$. The same function $f_\theta$ can be used to define an EBM for the joint distribution over $x$ and $y$ as: $p_\theta(x, y) = e^{f_\theta(x)[y]}/Z(\theta)$. The label $y$ can be marginalized out to give an unconditional model $p_\theta(x) = \sum_y e^{f_\theta(x)[y]}/Z(\theta)$. JEM models are trained to maximize the factorized likelihood:

$$\log p_\theta(x, y) = \alpha \log p_\theta(y|x) + \log p_\theta(x) \quad (11)$$

where $\alpha$ is a scalar which weights the two terms. The first term is optimized with cross-entropy and the second term is optimized using EBM training methods. In Grathwohl et al. (2019) PCD was used to train the second term. We train JEM models on CIFAR10, CIFAR100, and SVHN using VERA instead of PCD. We examine how this change impacts accuracy, generation, training speed, and stability. Full experimental details can be found in Appendix B.7.

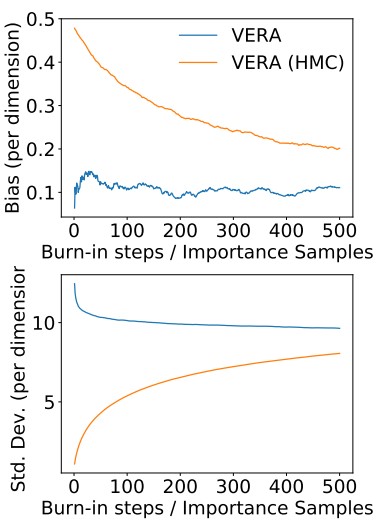

Figure 3: Bias (top) and standard deviation (bottom), both per dimension, of the score function estimator using HMC and our proposed importance sampling scheme.

**Speed and Stability**  While the results presented in Grathwohl et al. (2019) are promising, training models as presented in this work is challenging. MCMC sampling can be slow and training can easily diverge. Our CIFAR10 models train 2.8x faster than the official JEM implementation[3] with the default hyper-parameters. With these default parameters JEM models would regularly diverge. To train for the reported 200 epochs training needed to be restarted multiple times and the number of MCMC steps needed to be quadrupled, greatly increasing run-time.

Conversely, we find that VERA was much more stable and our models never diverged. This allowed us to remove the additive Gaussian noise added to the data which is very important to stabilize MCMC training (Grathwohl et al., 2019; Nijkamp et al., 2019a; Du & Mordatch, 2019).

**Hybrid Modeling**  In Tables 3 and 4 we compare the discriminative and generative performance of JEM models trained with VERA, PCD (JEM), and HDGE (Liu & Abbeel, 2020). With $\alpha = 1$ we find that VERA leads to models with poor classification performance but strong generation performance. With $\alpha = 100$ VERA obtains stronger classification performance than the original JEM model while still having improved image generation over JEM and HDGE (evaluated with FID (Heusel et al., 2017)).

---

[3]https://github.com/wgrathwohl/JEM

| Model | CIFAR10 | CIFAR100 | SVHN |
|---|---|---|---|
| Classifier | 95.8 | 74.2 | 97.7 |
| JEM | 92.9 | 72.2 | 96.6 |
| HDGE | 96.7 | 74.5 | N/A |
| VERA $\alpha = 100$ | 93.2 | 72.2 | 96.8 |
| VERA $\alpha = 1$ | 76.1 | 48.7 | 94.2 |

Table 3: Classification on image datasets.

| Model | FID ↓ |
|---|---|
| JEM | 38.4 |
| HDGE | 37.6 |
| SNGAN (Miyato et al., 2018) | 25.50 |
| NCSN (Song & Ermon, 2019b) | 23.52 |
| VERA $\alpha = 100$ | 30.5 |
| VERA $\alpha = 1$ | 27.5 |

Table 4: FID on CIFAR10.

Unconditional samples can be seen from our CIFAR10 and CIFAR100 models in Figure 4. Samples are refined through a simple iterative procedure using the latent space of our generator, explained in Appendix B.7.1. Additional conditional samples can be found in Appendix C.5

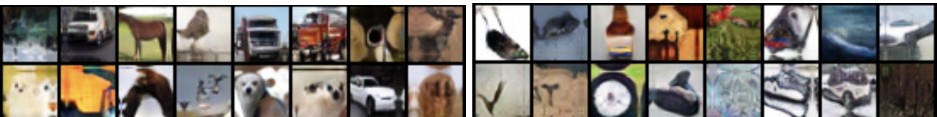

Figure 4: Unconditional samples on CIFAR10 (left) and CIFAR100 (right).

**Out-of-Distribution Detection** JEM is a powerful approach for out-of-distribution detection (OOD), greatly outperforming tractable likelihood models like VAEs and flows (Nalisnick et al., 2018). In Table 5, reporting AUROC (Hendrycks & Gimpel, 2016), we see that for all but 1 dataset, VERA outperforms JEM with PCD training but under-performs contrastive training (HDGE). Intriguingly, VERA performs poorly on CelebA. This result, along with the unreliable performance of DSM models at this task (Li et al., 2019) leads to questions regarding special benefits of MCMC training that are lost in our method as well as DSM. We leave this to future work.

| Model | SVHN | CIFAR100 | CIFAR10-Interp | CelebA |
|---|---|---|---|---|
| JEM | 0.67 | 0.67 | 0.65 | 0.75 |
| HDGE | 0.96 | 0.91 | 0.82 | 0.80 |
| GLOW | 0.05 | 0.51 | 0.55 | 0.57 |
| VERA | 0.83 | 0.73 | 0.86 | 0.33 |

Table 5: Out-of-distribution Detection. Model trained on CIFAR10. Values are AUROC (↑).

## 6.1 Tabular Data

Training with VERA is much more stable and easy to apply to domains beyond images where EBM training has been extensively tuned. To demonstrate this we show that JEM models trained with VERA can provide a benefit to semi-supervised classification on datasets from a variety of domains. Considerable progress has been made in semi-supervised learning but the most impressive results require considerable domain knowledge (Chen et al., 2020). In domains like images, text, and audio such knowledge exists but for data from particle accelerators, gyroscopes, and satellites, such intuition may not be available and these techniques cannot be applied. In these settings there are far fewer options for semi-supervised learning.

We present VERA as one such option. We train semi-supervised JEM models on data from a variety of continuous domains. We perform no data augmentation beyond removing redundant features and standardizing the remaining features. To further demonstrate the versatility of VERA we use an identical network for each dataset and method. Full experimental details can be found in Appendix B.8.

In Table 6, we find on each dataset tested, we find that VERA outperforms the supervised baseline and outperforms VAT which is the strongest domain agnostic semi-supervised learning method we are aware of.

## 7 Related Work

Kumar et al. (2019) train EBMs using entropy-regularized generators, attempting to optimize the same objective as our own. The key difference is how the generator's entropy is regularized. Kumar

---

[4]We treat MNIST as a tabular dataset since we do not use convolutional architectures.

| Model | HEPMASS | CROP | HUMAN | MNIST[4] |
|---|---|---|---|---|
| Supervised Baseline | 81.2 | 87.8 | 81.0 | 91.0 |
| VAT | 86.7 | 94.5 | 84.0 | 98.6 |
| MEG | 72.1 | 87.5 | 83.5 | 94.7 |
| JEM | 54.2 | 18.5 | 77.2 | 10.2 |
| VERA | 88.3 | 94.9 | 89.9 | 98.6 |
| Full-label | 90.9 | 99.7 | 98.0 | 99.5 |

Table 6: Accuracy of semi-supervised learning on tabular data with 10 labeled examples per class.

et al. (2019) utilize a Mutual Information estimator to approximate the generator's entropy whereas we approximate the gradients of the entropy directly. The method of Kumar et al. (2019) requires the training of an additional MI-estimation network, but our approach only requires the optimization of the posterior variance which has considerably fewer parameters. As demonstrated in Section 5.1, their approach does not perform as well as VERA for training NICE models and their generator collapses to a single point. This is likely due to the notorious difficulty of estimating MI in high dimensions and the unreliability of current approaches for this task (McAllester & Stratos, 2020; Song & Ermon, 2019a).

Dai et al. (2019) train EBMs using the same objective as VERA. The key difference here is in the architecture of the generator. Dai et al. (2019) use generators with a restricted form, inspired by various MCMC sampling methods. In this setting, a convenient estimator for the generator's entropy can be derived. In contrast, our generators have an unconstrained architecture and we focus on entropy regularization in the unconstrained setting.

Abbasnejad et al. (2019) train EBMs and generators as well to minimize the reverse KL-divergence. Their method differs from ours in the architecture of the generator and the method for encouraging high entropy. Their generator defines an implicit density (unlike ours which defines a latent-variable model). The entropy is maximized using a series approximation to the generator function's Jacobian log-determinant which approximates the change-of-variables for injective functions.

Gao et al. (2020) train EBMs using Noise Contrastive Estimation where the noise distribution is a normalizing flow. Their training objective differs from ours and their generator is restricted to having a normalizing flow architecture. These architectures do not scale as well as the GAN-style architectures we use to large image datasets.

As well there exist CoopNets (Xie et al., 2018) which cooperatively train an EBM and a generator network. Architecturally, they are similar to VERA but are trained quit differently. In CoopNets, the generator is trained via maximum likelihood on its own samples refined using MCMC on the EBM. This maximum likleihood step requires MCMC as well to generate posterior samples as in Pang et al. (2020a). In contrast, the generator in VERA is trained to minimize the reverse KL-divergence. Our method requires no MCMC and was specifically developed to alleviate the difficulties of MCMC sampling.

The estimator of Dieng et al. (2019) was very influential to our work. Their work focused on applications to GANs. Our estimator could easily be applied in this setting and to implicit variational inference (Titsias & Ruiz, 2019) as well but we leave this for future work.

## 8 CONCLUSION

In this work we have presented VERA, a simple and easy-to-tune approach for training unnormalized density models. We have demonstrated our approach learns high quality energy functions and models with high likelihood (when available for evaluation). We have further demonstrated the superior stability and speed of VERA compared to PCD training, enabling much faster training of JEM (Grathwohl et al., 2019) while retaining the performance of the original work. We have shown that VERA can train models from multiple data domains with no additional tuning. This enables the applications of JEM to semi-supervised classification on tabular data – outperforming a strong baseline method for this task and greatly outperforming JEM with PCD training.

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

Stephen Zhao, Jörn-Henrik Jacobsen, and Will Grathwohl. Joint energy-based models for semi-supervised classification.

## A    KEY DERIVATIONS

### A.1    DERIVATION OF VARIATIONAL LOG-PARTITION FUNCTION

Here we show that the variational optimization given in equation 3 recovers $\log Z(\theta)$.

$$\max_q \mathbb{E}_{q(x)}[f_\theta(x)] + H(q)$$
$$= \max_q \int_x q(x) f_\theta(x) \mathrm{d}x - \int_x q(x) \log(q(x)) \mathrm{d}x$$
$$= \max_q \int_x q(x) \log\left(\frac{\exp(f_\theta(x))}{q(x)}\right) \mathrm{d}x$$
$$= \max_q \int_x q(x) \log\left(\frac{\exp(f_\theta(x))}{q(x)}\right) \mathrm{d}x - \log Z(\theta) + \log Z(\theta)$$
$$= \max_q \int_x q(x) \log\left(\frac{\exp(f_\theta(x))/Z(\theta)}{q(x)}\right) \mathrm{d}x + \log Z(\theta)$$
$$= \max_q -\mathrm{KL}(q(x)\|p_\theta(x)) + \log Z(\theta)$$
$$= \log Z(\theta).$$

### A.2    SCORE FUNCTION ESTIMATOR

Here we derive the equivalent expression for $\nabla_x \log q_\phi(x)$ given in Equation 7.

$$\nabla_x \log q_\phi(x) = \frac{\nabla_x q_\phi(x)}{q_\phi(x)}$$
$$= \frac{\nabla_x \int_z q_\phi(x, z) dz}{q_\phi(x)}$$
$$= \frac{\int_z \nabla_x q_\phi(x, z) dz}{q_\phi(x)}$$
$$= \int_z \frac{\nabla_x q_\phi(x, z)}{q_\phi(x)} dz$$
$$= \int_z \frac{\nabla_x q_\phi(x \mid z) q_\phi(z)}{q_\phi(x)} dz$$
$$= \int_z \frac{(\nabla_x \log q_\phi(x \mid z)) q_\phi(x \mid z) q_\phi(z)}{q_\phi(x)} dz$$
$$= \mathbb{E}_{q_\phi(z|x)}[\nabla_x \log q_\phi(x|z)].$$

### A.3    ENTROPY GRADIENT ESTIMATOR

From Equation 6 we have

$$\nabla_\phi H(q_\phi) = \mathbb{E}_{p(z_0)p(\epsilon)}[\nabla_x \log q_\phi(x)^T \nabla_\phi x(z_0, \epsilon)].$$

Plugging in our score function estimator gives

$$
\begin{aligned}
\nabla_\phi H(q_\phi) &= \mathbb{E}_{p(z_0)p(\epsilon)}[\nabla_x \log q_\phi(x)^T \nabla_\phi x(z_0, \epsilon)] \qquad (12) \\
&= \mathbb{E}_{p(z_0)p(\epsilon)}\left[\mathbb{E}_{q_\phi(z|x)}\left[\nabla_x \log q_\phi(x \mid z)\right]^T \nabla_\phi x(z_0, \epsilon)\right] \\
&= \mathbb{E}_{p(z_0)p(\epsilon)}\left[\mathbb{E}_{\xi_\eta(z|z_0)}\left[\frac{q_\phi(z \mid x)}{\xi_\eta(z \mid z_0)}\nabla_x \log q_\phi(x \mid z)\right]^T \nabla_\phi x(z_0, \epsilon)\right] \\
&= \mathbb{E}_{p(z_0)p(\epsilon)}\left[\mathbb{E}_{\xi_\eta(z|z_0)}\left[\frac{q_\phi(x, z)}{q_\phi(x)\xi_\eta(z \mid z_0)}\nabla_x \log q_\phi(x \mid z)\right]^T \nabla_\phi x(z_0, \epsilon)\right] \\
&\approx \mathbb{E}_{p(z_0)p(\epsilon)}\left[\left[\sum_{i=1}^{k}\frac{w_i}{\sum_{j=1}^{k} w_j}\nabla_x \log q_\phi(x \mid z_i)\right]^T \nabla_\phi x(z_0, \epsilon)\right]
\end{aligned}
$$

where $\{z_i\}_{i=1}^{k} \sim \xi(z \mid z_0)$ and $w_i \equiv \frac{q_\phi(z_i, x)}{\xi(z_i|z_0)}$.

### A.3.1 DISCUSSION

We discuss when the approximations in Equation 13 will hold. The importance sampling estimator will be biased when $q_\psi(z \mid x)$ differs greatly from $\xi(z \mid z_0)$. Since the generator function $g_\psi(z)$ is a smooth, Lipschitz function (as are most neural networks) and the output Gaussian noise is small, the space of $z$ values which could have generated $x$ should be concentrated near $z_0$. In these settings, $z_0$ should be useful for predicting $q(z|x)$.

The accuracy of this approximation is based on the dimension of $z$ compared to $x$ and the Lipschitz constant of $g_\psi$. In all settings we tested, $\dim(z) \ll \dim(x)$ where this approximation should hold. If $\dim(z) \gg \dim(x)$ the curse of dimensionality would take effect and $z_0$ would be less and less informative about $q(z|x)$. In settings such as these, we do not believe our approach would be as effective. Thankfully, almost all generator architectures we are aware of have $\dim(z) \ll \dim(x)$. The approximation could also break down if the Lipschitz constant blew up. We find this does not happen in practice, but this can be addressed with many forms of regularization and normalization.

# B    EXPERIMENTAL DETAILS

## B.1    HYPERPARAMETER RECOMMENDATIONS

| Type | Hyperparameter | Value |
|------|----------------|-------|
| Optimization | $\beta_1$ (ADAM) | 0 |
| | $\beta_2$ (ADAM) | 0.9 |
| | learning rate (energy) | $10^{-4*}$ |
| | learning rate (generator) | $2 \cdot 10^{-4*}$ |
| | learning rate (posterior) | $2 \cdot 10^{-4*}$ |
| Regularization | $\lambda$ (entropy regularization weight) | $10^{-4\dagger}$ |
| | $\gamma$ (gradient norm penalty) | 0.1 |
| JEM | $\alpha$ (classification weight) | $\{1, 10, 30, 100\}^*$ |
| | $\beta$ (classification entropy) | $\{1, 0.1 * \alpha, \alpha\}$ |

Table 7: Hyperparameters for VERA.

*When $\alpha > 1$, learning rates were divided by $\alpha$.

†We found $\lambda = 10^{-4}$ to work best on large image datasets, but in general we recommend starting with $\lambda = 1$ and trying successively smaller values of $\lambda$ until training is stable.

We give some general tips on how to set hyperparameters when training VERA in Table 7. In all VERA experiments, we use the gradient norm penalty with weight 0.1. This was not tuned during our experiments. When using VERA and MEG we train with Adam (Kingma & Ba, 2014) and set $\beta_1 = 0, \beta_2 = .9$ as is standard in the GAN literature (Miyato et al., 2018). In general, we recommend setting the learning rate for the generator to twice the learning rate of the energy function and equal to the learning rate of the approximate posterior sampler.

### B.1.1    IMPACT OF $\lambda$

Let us rewrite the generator's training objective

$$\mathcal{L}(q; \lambda) = E_{q(x)}[f_\theta(x)] + \lambda H(q). \tag{13}$$

We can easily see that this objective is equivalent (up to a multiplicative constant) to

$$E_{q(x)} \left[ \frac{f_\theta(x)}{\lambda} \right] + H(q). \tag{14}$$

From this, it is clear that maximizing Equation 14 is the same as minimizing the KL-divergence between $q$ and a tempered version of $p_\theta(x)$ defined as

$$\frac{e^{f_\theta(x)/\lambda}}{Z}. \tag{15}$$

Tempering like this is standard practice in EBM training and is done in many recent works. Tempering has the effect of increasing the weight of the gradient signal in SGLD sampling relative to the added Gaussian noise. In all of Du & Mordatch (2019); Grathwohl et al. (2019); Nijkamp et al. (2019a;b) the SGLD samplers used a temperature of $1/\lambda = 20,000$. This value is near the value of $10,000$ that we use in this work.

Thus we can see that our most important hyper-parameter is actually the temperature of the sampler's target distribution. Ideally this temperature would be set to 1, but this can lead to unstable training (as it does with SGLD). To train high quality models, we recommend setting $\lambda$ as close to 1 as possible, decreasing if training becomes unstable.

## B.2    TOY DATA VISUALIZATIONS

We train simple energy functions on a 2D toy data. The EBM is a 2-layer MLP with 100 hidden units per layer using the Leaky-ReLU nonlinearity with negative slope .2. The generator is a 2-layer

MLP with 100 hidden units per layer and uses batch normalization (Ioffe & Szegedy, 2015) with ReLU nonlinearities. All models were trained for 100,000 iterations with all learning rates set to .001 and used the Adam optimizer (Kingma & Ba, 2014).

The PCD models were trained using an SGLD sampler and a replay buffer with 10,000 examples, reinitialized every iteration with $5\%$ probability. We used 20 steps of SGLD per training iteration to make runtime consistent with VERA. We tested $\sigma$ values outside of the presented range but smaller values did not produce decent samples or energy functions and for larger values, training diverged.

### B.3 TRAINING NICE MODELS

The NICE models were exactly as in Song et al. (2020). They have 4 coupling layers and each coupling layer had 5 hidden layers. Each hidden layer has 1000 units and uses the Softplus nonlinearity. We preprocessed the data as in Song et al. (2020) by scaling the data to the range $[0, 1]$, adding uniform noise in the range $[-1/512, 1/512]$, clipping to the range $[.001, .999]$ and applying the logit transform $\log(x) - \log(1 - x)$. All models were trained for 400 epochs with the Adam optimizer (Kingma & Ba, 2014) with $\beta_1 = 0$ and $\beta_2 = .9$. We use a batch size of 128 for all models. We re-ran the score matching model of Song et al. (2020) to train for 400 epochs as well and found it did not improve performance as its best test performance happens very early in training.

For all generator-based training methods we use the same fixed generator architecture. The generator has a latent-dimension of 100 and 2 hidden layers with 500 units each. We use the Softplus nonlinearity and batch-normalization (Ioffe & Szegedy, 2015) as is common with generator networks.

For VERA the hyper-parameters we searched over were the learning rates for the NICE model and for the generator. Compared to Song et al. (2020) we needed to use much lower learning rates. We searched over learning rates in $\{.0003, .00003, .000003\}$ for both the generator and energy function. We found .000003 to work best for the energy function and .0003 to work best for the generator. This makes intuitive sense since the generator needs to be fully optimized for the bound on likelihood to be tight. When equal learning rates were used (.0003, .0003) we observed high sample quality from the *generator* but exact samples and likelihoods from the NICE model were very poor.

For PCD we search over learning rates in $\{.0003, .00003, .000003\}$, the number of MCMC steps in $\{20, 40\}$ and the SGLD noise standard-deviation in $\{1.0, 0.1\}$. All models with 20 steps and SGLD standard-deviation 1.0 quickly diverged. Our best model used learning rate .000003, stepsize 0.1, and 40 steps. We tested the gradient-norm regularizer from Kumar et al. (2019) and found it decreased performance for PCD trained models. Most models with 20 MCMC steps diverged early in training.

For review the MCMC sampler we use is stochastic gradient Langevin dynamics (Welling & Teh, 2011). This sampler updates its samples by

$$x_t = x_{t-1} + \frac{\sigma^2}{2} \nabla_x f_\theta(x) + \epsilon \sigma, \qquad \epsilon \sim \mathcal{N}(0, I) \tag{16}$$

where $\sigma$ is the noise standard-deviation and is a parameter of the sampler.

For Maximum Entropy Generators (MEG) (Kumar et al., 2019) we must choose a mutual-information estimation network. We follow their work and use an MLP with LeakyReLU non-linearities with negative slope .2. Our network mirrors the generator and has 3 hidden layers with 500 units each. We searched over the same hyper-parameters as VERA. We found MEG to perform almost identically to training with no entropy regularization at all. We believe this has to do with the challenges of estimating MI in high dimensions Song & Ermon (2019a).

For CoopNets (Xie et al., 2018) we use the same flow and generator architectures as VERA. Following the MNIST experiments in Xie et al. (2018) we train using 10 SGLD steps per iteration. We tried the recommended learning rates of .007 and .0001 for the flow and generator, respectively but found this to lead quick divergence. For this reason, we search over learning rates for the flow and generator from $\{.0003, .00003, .000003\}$ as we did for VERA and found the best combination to be .000003 for the flow and .0003 for the generator. Other combinations resulted in higher quality generator samples but *much* worse likelihood values. We tested the recommended SGLD step-size of .002 and found this to lead to divergence as well in this setup. Thus, we searched over larger values of $\{.002, .02, .1\}$ found .1 to perform the best, as with PCD.

### B.4    ESTIMATION OF BIAS OF ENTROPY REGULARIZER

If we restrict the form of our generator to a linear function

$$x = Wz + \mu + \sigma\epsilon, \qquad z, \epsilon \sim \mathcal{N}(0, I)$$

then we have

$$q(x|z) = \mathcal{N}(Wz + \mu, \sigma^2 I), \qquad q(x) = \mathcal{N}(\mu, W^T W + \sigma^2 I)$$

meaning we can exactly compute $\log q(x)$, and $\nabla_x \log q(x)$ which is the quantity that VERA (HMC) approximates with the HMC estimator from Dieng et al. (2019) and we approximate with VERA. To explore this, we fit a PCA model on MNIST and recover the parameters $W, \mu$ of the linear generator and the noise parameter $\sigma$. Samples from this model can be seen in Figure 5.

Both VERA and VERA (HMC) have some parameters which are tuned automatically to improve the estimator throughout training. For VERA this is the posterior variance which is optimized according to Equation 10 with Adam with default hyperparameters. For VERA (HMC) this is the stepsize of HMC which is tuned automatically as outlined in (Dieng et al., 2019). Both estimators were trained with a learning rate of .01 for 500 iterations with a batch size of 5000. Samples from the estimators during training were taken with the default parameters, in particular the number of burn-in steps or the number of importance samples was not varied as they were during evaluation.

The bias of the estimators were evaluated on a batch of 10 samples from the generator. For each example in the batch, 5000 estimates were taken from the estimator and averaged to be taken as an estimate of the score function for this batch example. This estimate of the score function was subtracted from the true score function and then averaged over all dimensions and examples in the batch and taken as an estimate of the bias per dimension.

For VERA (HMC) we varied the number of burn-in steps used for samples to evaluate the bias of the estimator. We also tried to increase the number of posterior samples taken off the chain, but we found that this did not clearly reduce the bias of this estimator as the number of samples increased. For VERA we computed the bias on the default of 20 importance samples.

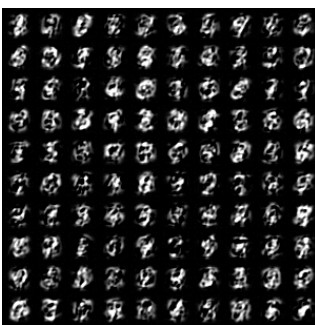

Figure 5: Samples from linear model trained with PCA on MNIST.

### B.5    MODE COUNTING

We train VERA without labels on the STACKEDMNIST dataset. This dataset consists of 60,000 examples of 3 stacked MNIST images sampled with replacement from the original 60,000 MNIST train set. As in (Dieng et al., 2019) we resize the images to $64 \times 64$ and use the DCGAN (Radford et al., 2015) architecture for both the energy-function and generator. We use a latent code of 100 dimensions. We train for 17 epochs with a learning rate of .001 and batch size 100.

We estimate the number of modes captured by taking $S = 10,000$ samples as in (Dieng et al., 2019) and classifying each digit of the 3 stacked images separately with a pre-trained classifier on MNIST.

### B.6    EFFECTIVE SAMPLE SIZE

When performing importance sampling, the quality of the proposal distribution has a large impact. If the proposal is chosen poorly, then typically 1 sample will dominate in the expectation. This can

be quantified using the effective sample size (Kong, 1992) (ESS) which is defined as

$$w_i = \frac{p(x)}{q(x)}$$

$$\tilde{w}_i = \frac{w_i}{\sum_{j=1}^{N} w_j}$$

$$\text{ESS} = \frac{1}{\sum_{i=1}^{N} \tilde{w}_i^2} \qquad (17)$$

where $p(x)$ is the target distribution, $q(x)$ is the proposal distribution and $N$ is the number of samples. If the self-normalized importance weights are dominated by one weight close to 1 then the ESS will be 1. If the proposal distribution is identical to the target, so that the self-normalized importance weights are then uniform, then the ESS will be $N$. When $\text{ESS} = N$, importance sampling is as efficient as using the target distribution.

To understand the effect of the proposal distribution on ESS we plot the ESS when doing importance sampling with 20 importance samples from a 128-dimensional Gaussian target distribution with $\mu = 0$ and $\Sigma = I$. We use a proposal which is a 128-dimensional Gaussian with $\mu$ increasing from 0 to 5. Results can be seen in Figure 6. We see when the means differ by greater than 2, the ESS is approximately 1.0 and importance sampling has effectively failed.

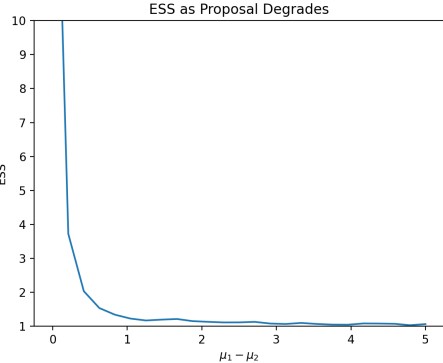

Figure 6: Effective sample size of importance sampling as the proposal degrades.

## B.7 JEM MODELS

Our energy function used the same architecture as in Grathwohl et al. (2019); Liu & Abbeel (2020), a Wide ResNet Zagoruyko & Komodakis (2016) 28-10 with batch-norm removed and with dropout. Our generator architecture is identical to Miyato et al. (2018). As in Grathwohl et al. (2019); Liu & Abbeel (2020) we set the learning rate for the energy function equal to 0.0001. We set the learning rate for the generator equal to 0.0002. We train for 200 epochs using the Adam optimizer with $\beta_1 = 0$ and $\beta_2 = .9$. We set the batch size to 64. Results presented are from the models after 200 epochs with no early stopping. We believe better results could be obtained from further training.

We trained models with $\alpha \in \{1, 30, 100\}$ and found classification to be best with $\alpha = 100$ and generation to be best with $\alpha = 1$.

Prior work on PCD EBM training (Grathwohl et al., 2019; Du & Mordatch, 2019; Nijkamp et al., 2019a;b) recommends adding Gaussian noise to the data to stabilize training. Without this, PCD training of JEM models very quickly diverges. Early in our experiments we found training with VERA was stable without the addition of Gaussian noise so we do not use it.

As mentioned in Dieng et al. (2019), when the strength of the entropy regularizer $\lambda$ is too high, the generator may fall into a degenerate optimum where it just outputs high-entropy Gaussian noise. To combat this, as suggested in Dieng et al. (2019), we decrease the strength of $\lambda$ to .0001 for all JEM experiments. This value was chosen by decreasing $\lambda$ from 1.0 by factors of 10 until learning took place (quantified by classification accuracy).

### B.7.1 MCMC Sample Refinement

Our generator $q_\phi(x)$ is trained to approximate our EBM $p_\theta(x)$. After training, the samples from the generator are of high quality (see Figures 9 and 10, left) but they are not exactly samples from $p_\theta(x)$. We can use MCMC sampling to improve the quality of these samples. We use a simple MCMC refinement procedure based on the Metropolis Adjusted Langevin Algorithm (Besag, 1994) applied to an expanded state-space defined by our generator and perform the Accept/Reject step in the data space.

We can reparameterize a generator sample $x \sim q_\phi(x)$ as a function $x(z, \epsilon) = g_\psi(z) + \epsilon\sigma$, and we can define an unnormalized density over $\{z, \epsilon\}$, $\log h(z, \epsilon) \equiv f_\theta(g_\psi(z) + \epsilon\sigma) - \log Z(\theta, \phi)$ which is the density (under $p_\theta(x)$) of the generator sample.

Starting with an initial sample $z_0, \epsilon_0 \sim \mathcal{N}(0, I)$ we define the proposal distribution

$$p(z_t|z_{t-1}, \epsilon_{t-1}) = \mathcal{N}\left(z_{t-1} + \frac{\delta}{2}\nabla_{z_{t-1}} \log h(z_{t-1}, \epsilon_{t-1}), \delta^2 I\right)$$

$$p(\epsilon_t|z_{t-1}, \epsilon_{t-1}) = \mathcal{N}\left(z_{t-1} + \frac{\delta}{2}\nabla_{\epsilon_{t-1}} \log h(z_{t-1}, \epsilon_{t-1}), \delta^2 I\right)$$

$$p(z_t, \epsilon_t|z_{t-1}, \epsilon_{t-1}) = p(z_t|z_{t-1}, \epsilon_{t-1})p(\epsilon_t|z_{t-1}, \epsilon_{t-1})$$

and accept a new sample with probability

$$\min\left[\frac{h(z_t, \epsilon_t)p(z_t, \epsilon_t|z_{t-1}, \epsilon_{t-1})}{h(z_{t-1}, \epsilon_t)p(z_{t-1}, \epsilon_{t-1}|z_t, \epsilon_t)}, 1\right].$$

Here, $\delta$ is the step-size and is a parameter of the sampler. We tune $\delta$ with a burn-in period to target an acceptance rate of $0.57$.

A similar sampling procedure was proposed in Kumar et al. (2019) and Che et al. (2020) and in both works was found to improve sample quality. In all experiments,

We clarify that this procedure is *not* a valid MCMC sampler for $p_\theta(x)$ due to the augmented variables and the change in density of $g_\psi$ which are not corrected for. The density of the samples will be a combination of $p_\theta(x)$ and $q_\phi(x)$. As the focus of this work was training and not sampling/generation, we leave the development of more correct generator MCMC-sampling to future work. Regardless, we find this procedure to improve visual sample quality. In Figure 7 we visualize a sampling chain using the above method applied to our JEM model trained on SVHN.

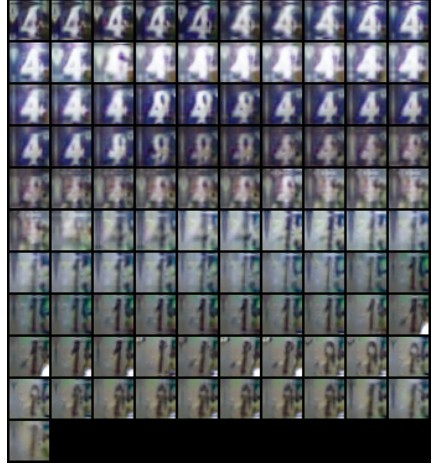 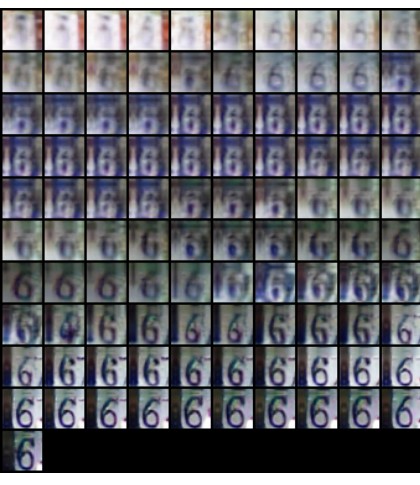

Figure 7: Visualization of our MALA-inspired sample refinement procedure. Samples come from JEM model trained on SVHN. Chains progress to the right and down. Each image is a consecutive step, no sub-sampling is done.

### B.7.2 IMAGE QUALITY METRICS

We present results on Inception Score (Salimans et al., 2016) and Frechet Inception Distance (Heusel et al., 2017). These metrics are notoriously fickle and different repositories are known to give very different results (Grathwohl et al., 2019). For these evaluations we generate 12,800 samples from the model and (unless otherwise stated) refine the samples with 100 steps of our latent-space MALA procedure (Appendix B.7.1). The code to generate our reported FID comes from this publicly available repository. The code to generate our reported Inception Score can be found here.

### B.8 SEMI-SUPERVISED LEARNING ON TABULAR DATA

### B.8.1 DATA

We provide details about each of the datasets used for the experiments in Section 6.1. HEPMASS[5] is a dataset obtained from a particle accelerator where we must classify signal from background noise. CROP[6] is a dataseset for classifying crop types from optical and radar sensor data. HUMAN[7] is a dataset for human activity classification from gyroscope data. MNIST is an image dataset of handwritten images, treated here as tabular data.

| Dataset | HEPMASS | CROP | HUMAN | MNIST |
|---|---|---|---|---|
| Features | 15 | 174 | 523 | 784 |
| Examples | 6,300,000 | 263,926 | 6,617 | 60,000 |
| Classes | 2 | 7 | 6 | 10 |
| Max class % | 50.0 | 26.1 | 19.4 | 11.2 |

Table 8: Basic information about each tabular dataset.

For HEPMASS and HUMAN data we remove features which repeat the same exact same value more than 5 times. For CROP data we remove features which have covariation greater than 1.01. For MNIST we linearly standardize features to the interval $[-1, 1]$.

We take a random 10% subset of the data to use as a validation set.

### B.8.2 TRAINING

We use the same architecture for all experiments and baselines. It has 6 layers of hidden units with dimensions $[1000, 500, 500, 250, 250, 250]$ and a Leaky-ReLU nonlinearity with negative-slope .2 between each layer of hidden units. The only layers which change between datasets are the input layer and the output layer which change according to the number of features and number of classses respectively.

The training process for semi-supervised learning is similar to JEM with an additional objective commonly used in semi-supervised learning:

$$\log p_\theta(x, y) = \alpha \log p_\theta(y \mid x) + \log p_\theta(x) + \beta H(p_\theta(y \mid x)) \tag{18}$$

where $H(p(y|x))$ is the entropy of the predictive distribution over the labels.

For all models we report the accuracy the model converged to on the held-out validation set. We report the average taken over three training runs with different seeds.

We use equal learning rates for the energy model, generator, and the entropy estimator. We tune the learning rate and decay schedule for supervised models on the full-set of labels and 10 labels per class.

---

[5] http://archive.ics.uci.edu/ml/datasets/HEPMASS
[6] https://archive.ics.uci.edu/ml/datasets/Crop+mapping+using+fused+optical-radar+data+set
[7] https://archive.ics.uci.edu/ml/datasets/Human+Activity+Recognition+Using+Smartphones

On VERA we tune the learning rate and decay schedule, the weighting of the entropy regularization $\lambda$ and the weighting of the entropy of classification outputs $\beta$.

For VAT we tune the perturbation size $\epsilon \in \{.01, .1, 1, 3, 10\}$. All other hyperparameters were fixed according to tuning on VERA.

For MEG we used the hyperparameters tuned according to VERA.

For JEM we tune the number of MCMC steps in the range $\kappa \in \{20, 40, 80\}$. We generate samples using SGLD with stepsize 1 and noise standard deviation 0.01 as in Grathwohl et al. (2019).

| Dataset | HEPMASS | CROP | HUMAN | MNIST |
|---|---|---|---|---|
| Learning rate | $10^{-5}$ | $10^{-4}$ | $10^{-5}$ | $10^{-5}$ |
| Number of epochs | 1 | 20 | 800 | 200 |
| Batch size | 64 | 64 | 64 | 64 |
| Decay rate | .3 | .3 | .3 | .3 |
| Decay epochs | $\{.05, .1\}$ | $\{15, 17\}$ | $\{200, 300, 400, 600\}$ | $\{150, 175\}$ |
| $z$ (latent) dimension | 10 | 128 | 128 | 128 |
| $\lambda$ | $10^{-4}$ | $10^{-4}$ | $10^{-4}$ | $10^{-4}$ |
| $\alpha$ | 1 | 1 | 10 | 1 |
| $\beta$ | 1 | 1 | 1 | 1 |
| $\epsilon$ (VAT) | .1 | 3 | .1 | 3 |
| $\kappa$ (JEM) | 20 | 20 | 40 | 80 |

Table 9: Hyperparameters for semi-supervised learning on tabular data

# C  ADDITIONAL RESULTS

## C.1  TRAINING MIXTURES OF GAUSSIANS

We present additional results training mixure of Gaussian models using VERA and PCD. Each model consits of 100 diagonal Gaussians and mixing weights. Our experimental setup and hyper-parameter search was identical to that presented in Appendiex B.2. We see in Table 10 that VERA outperforms PCD.

|  | Max. Likelihood | VERA | | | | PCD |
|---|---|---|---|---|---|---|
|  |  | $\lambda = 0.0$ | $\lambda = .01$ | $\lambda = 0.1$ | $\lambda = 1.0$ |  |
| Moons | -2.32 | -3.87 | -3.63 | -3.10 | **-2.58** | -3.82 |
| Circles | -3.17 | -3.58 | -3.64 | -3.74 | **-3.40** | -4.24 |
| Rings | -2.83 | -3.5 | -3.44 | -3.24 | **-3.17** | -4.13 |

Table 10: Fitting a mixture of 100 diagonal Gaussians using ML, MCMC approximate ML and generator approximate ML.

## C.2  SAMPLES FROM SSL MODELS

We present some samples from our semi-supervised MNIST models in Figure 8.

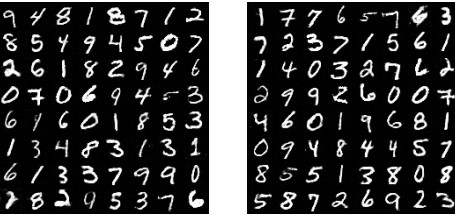

Figure 8: Unconditional MNIST Samples. Left: samples from the generator and right: samples after 100 steps if MCMC refinement using MALA.

## C.3  HYBRID MODELING

We present an extended Table 4 with Inception Score Salimans et al. (2016) and which includes more comparisons.

| Model | FID | IS |
|---|---|---|
| JEM | 38.4 | 8.76 |
| HDGE | 37.6 | 9.19 |
| SNGAN (Miyato et al., 2018) | 25.50 | 8.59 |
| NCSN (Song & Ermon, 2019b) | 23.52 | 8.91 |
| ADE (Dai et al., 2019) | N/A | 7.55 |
| IGEBM (Du & Mordatch, 2019) | 37.9 | 8.30 |
| Glow (Kingma & Dhariwal, 2018) | 48.9 | 3.92 |
| FCE (Gao et al., 2020) | 37.3 | N/A |
| VERA $\alpha = 100$ | 30.5 | 8.11 |
| VERA $\alpha = 1$ | 27.5 | 8.00 |
| VERA $\alpha = 1$ (generator samples) | 32.4 | 7.34 |

## C.4 UNCONDITIONAL IMAGE GENERATION: CIFAR10

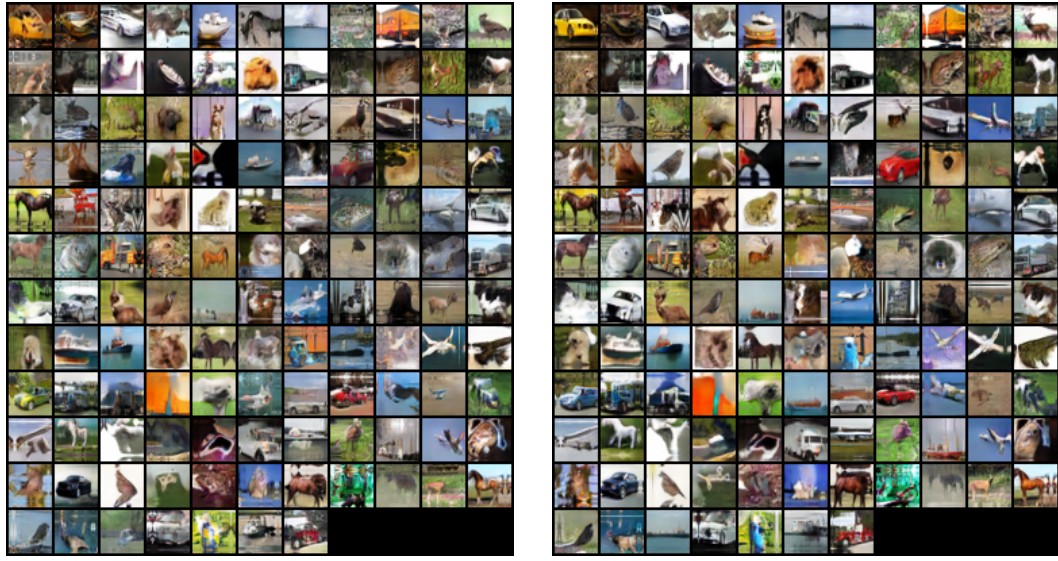

Figure 9: Unconditional CIFAR10 Samples. Left: samples from the generator and right: samples after 100 steps if MCMC refinement using MALA.

## C.5 UNCONDITIONAL IMAGE GENERATION: CIFAR100

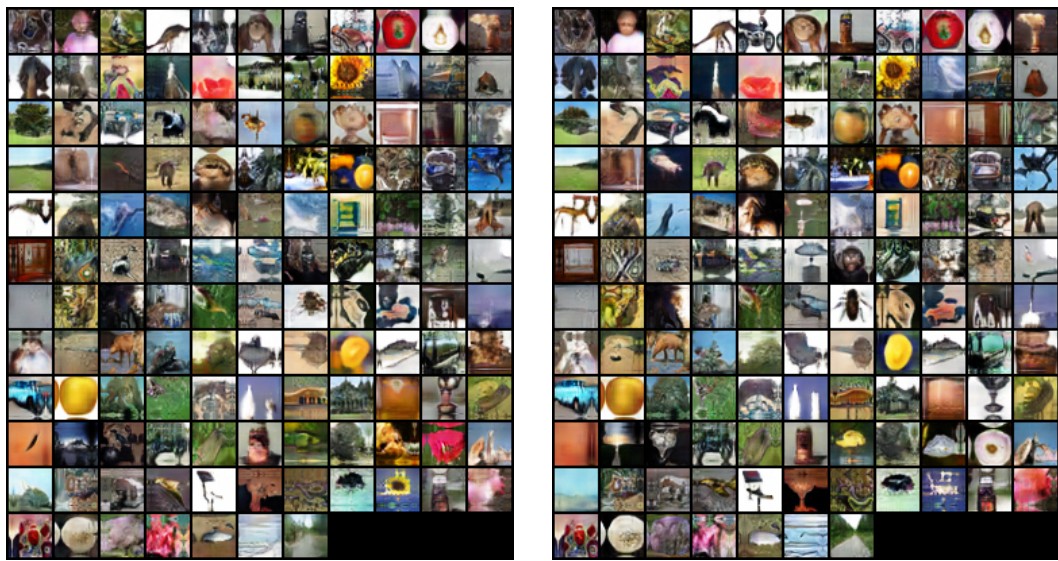

Figure 10: Unconditional CIFAR100 Samples. Left: samples from the generator and right: samples after 100 steps if MCMC refinement using MALA.

## C.6 Conditional Image Generation: CIFAR10 and CIFAR100

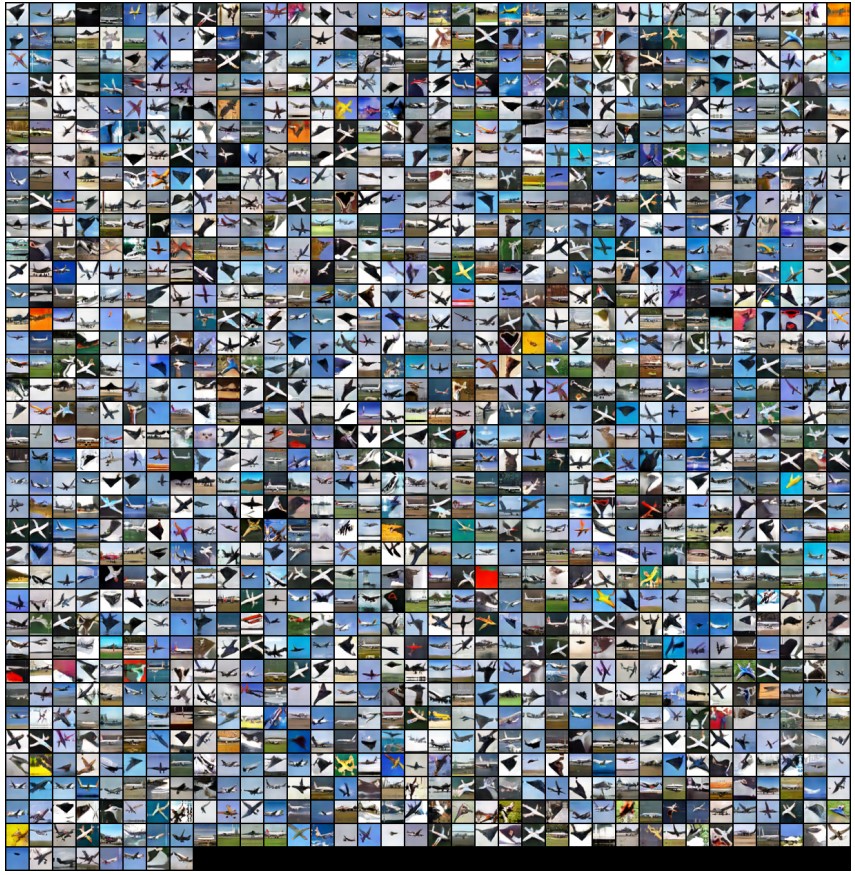

Figure 11: Class-conditional samples from CIFAR10

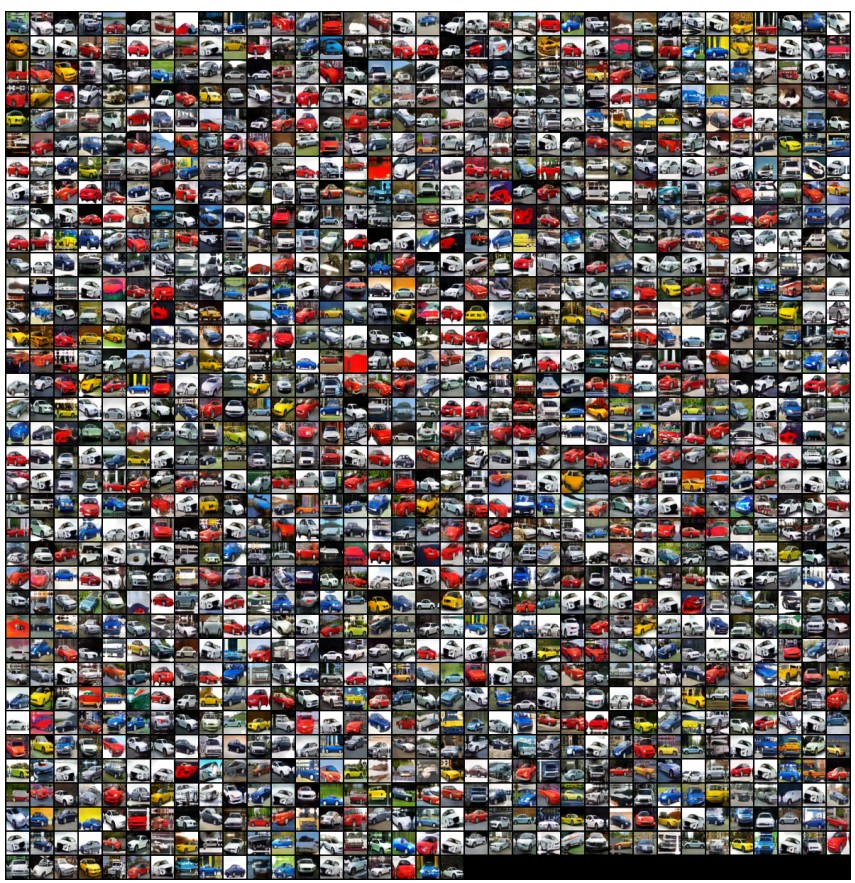

Figure 12: Class-conditional samples from CIFAR10

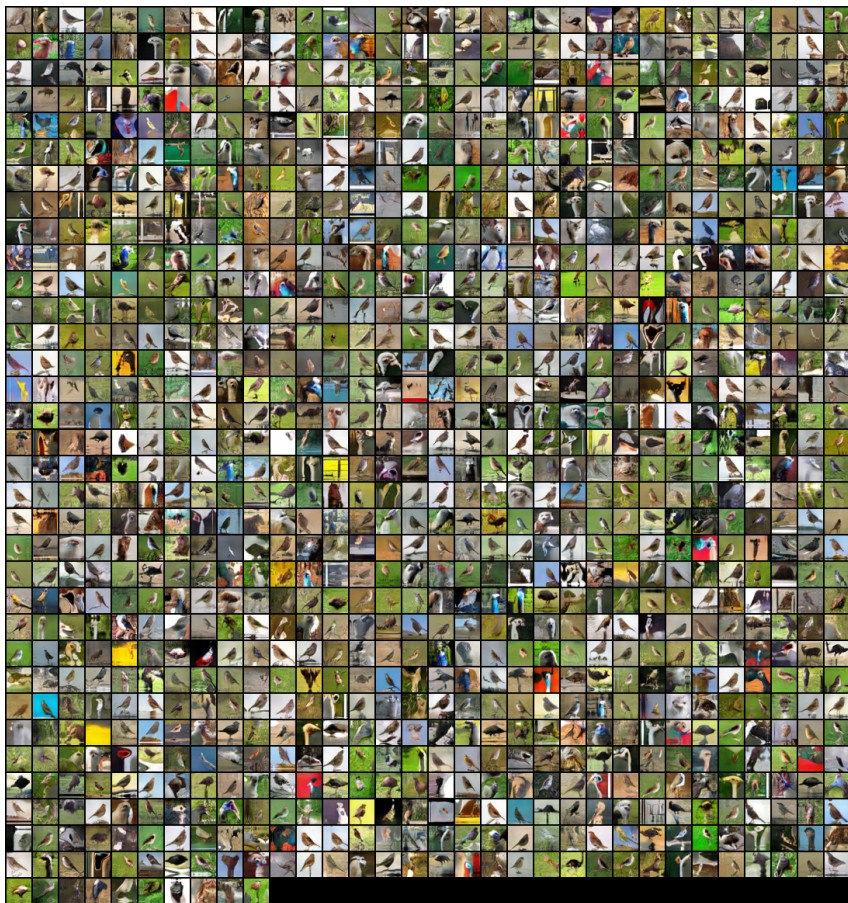

Figure 13: Class-conditional samples from CIFAR10

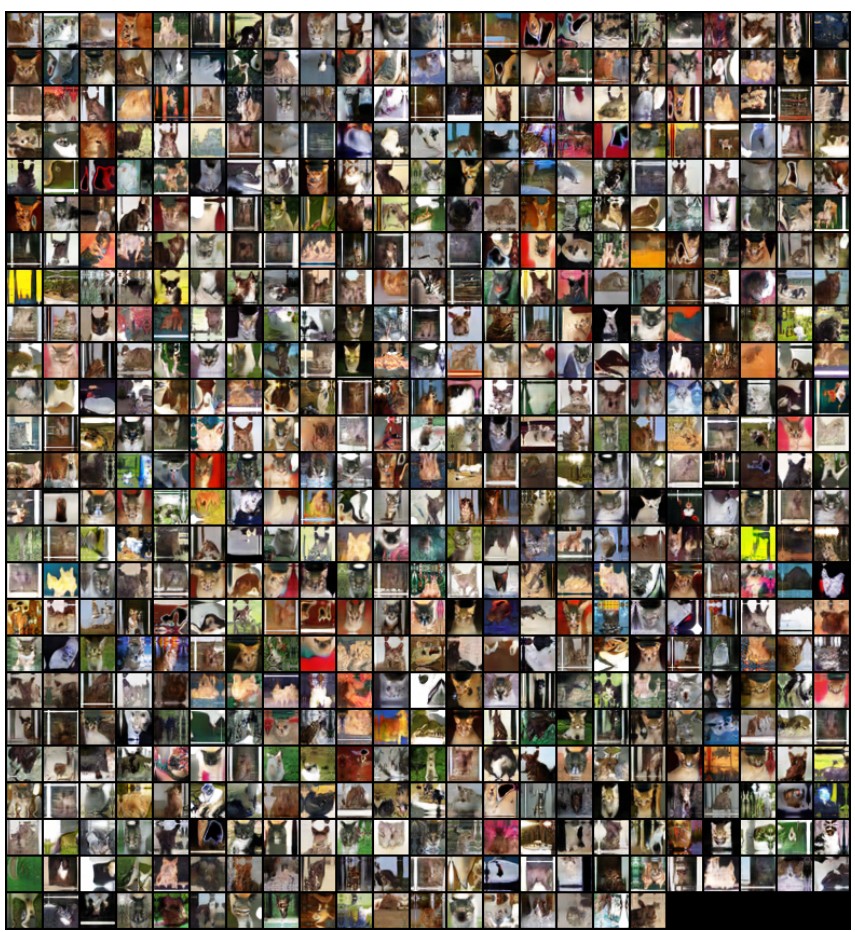

Figure 14: Class-conditional samples from CIFAR10

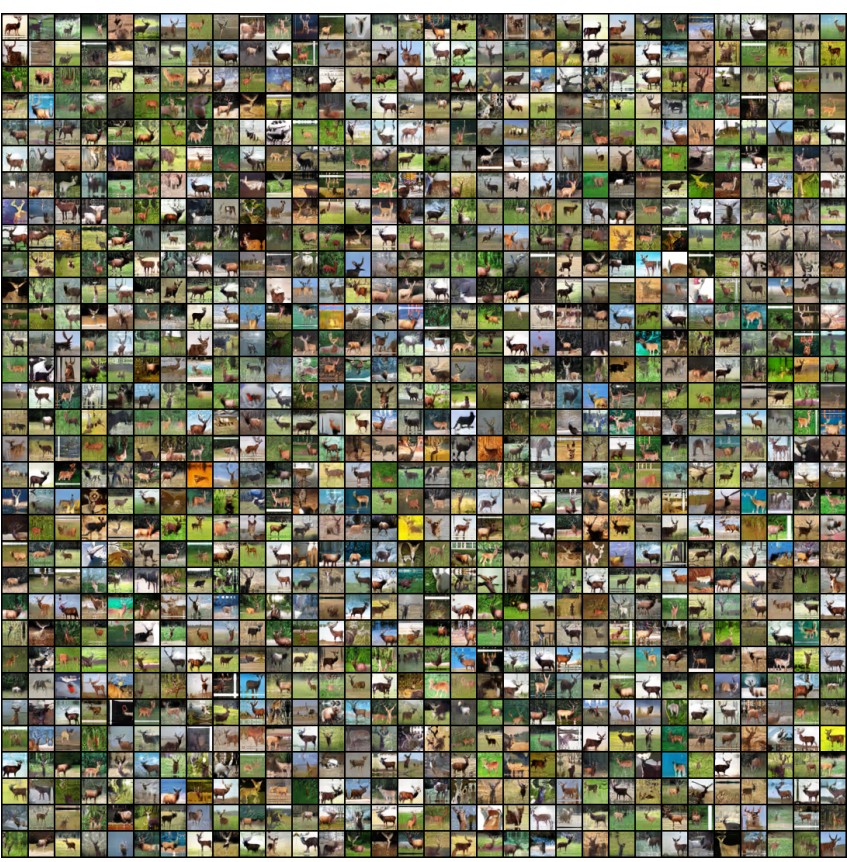

Figure 15: Class-conditional samples from CIFAR10

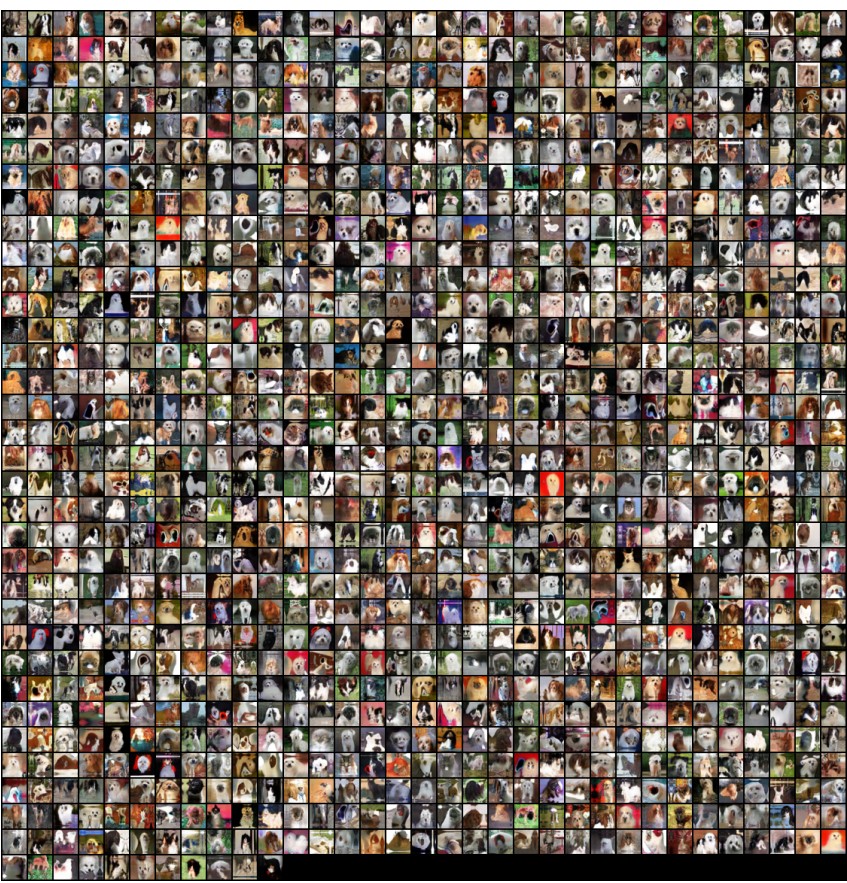

Figure 16: Class-conditional samples from CIFAR10

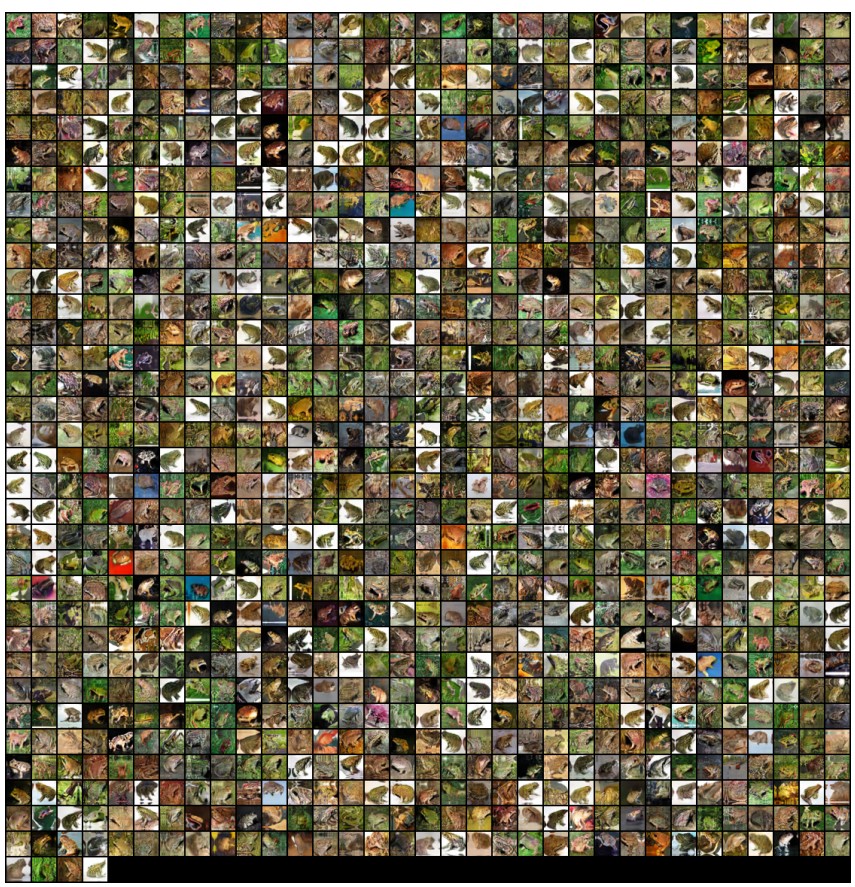

Figure 17: Class-conditional samples from CIFAR10

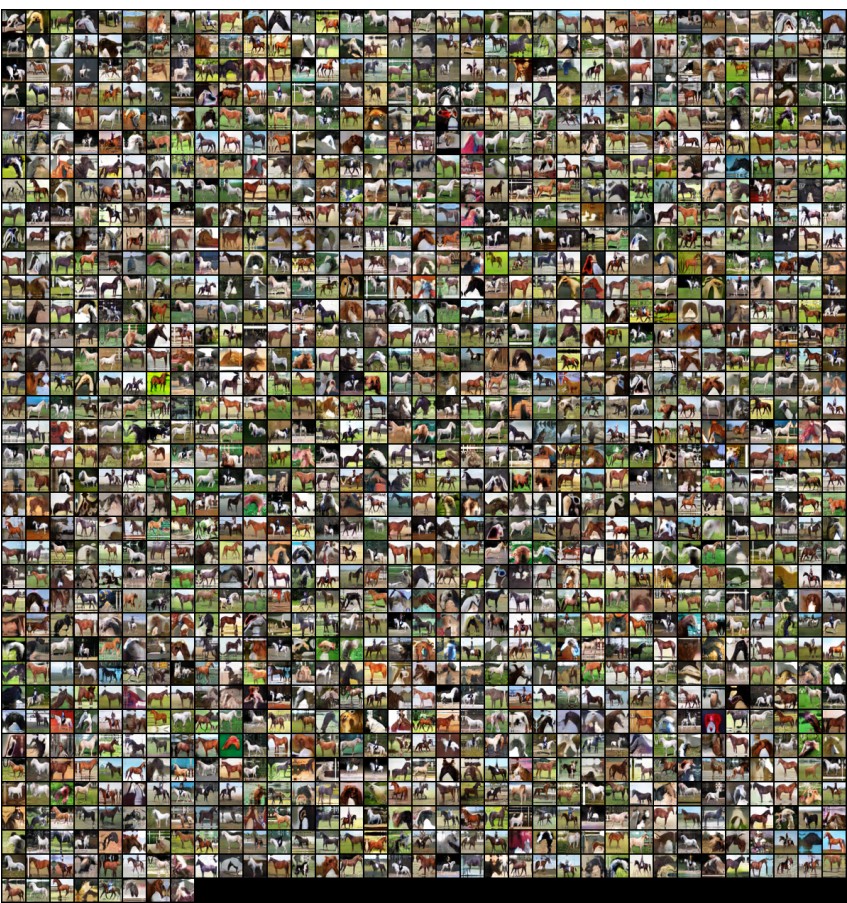

Figure 18: Class-conditional samples from CIFAR10

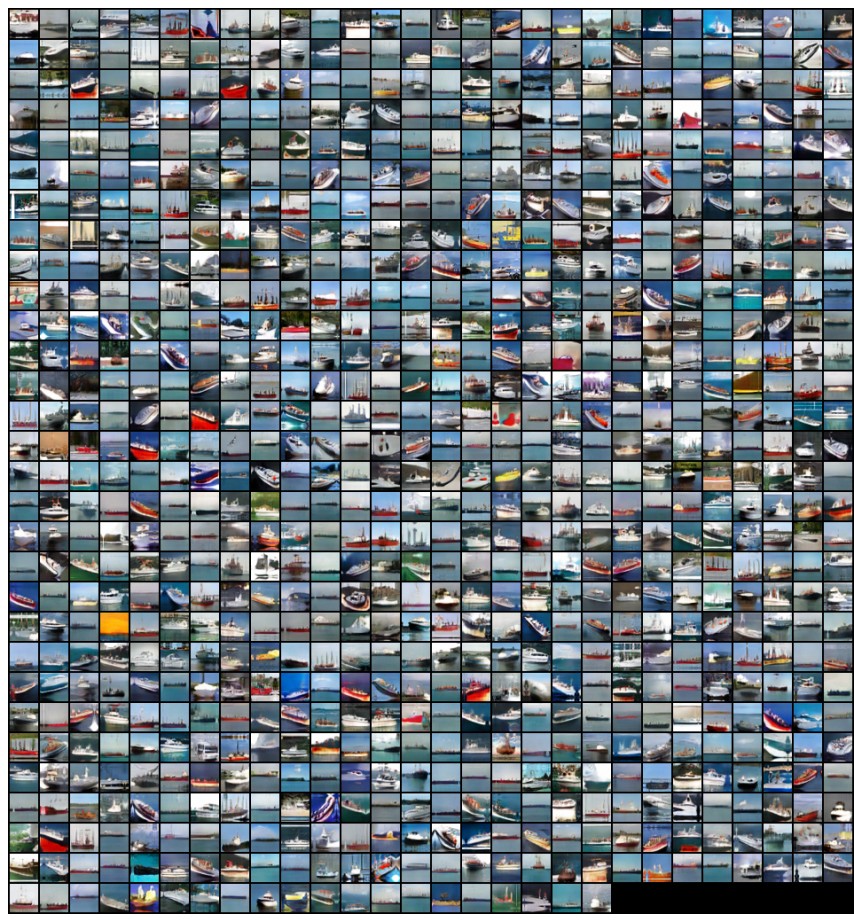

Figure 19: Class-conditional samples from CIFAR10

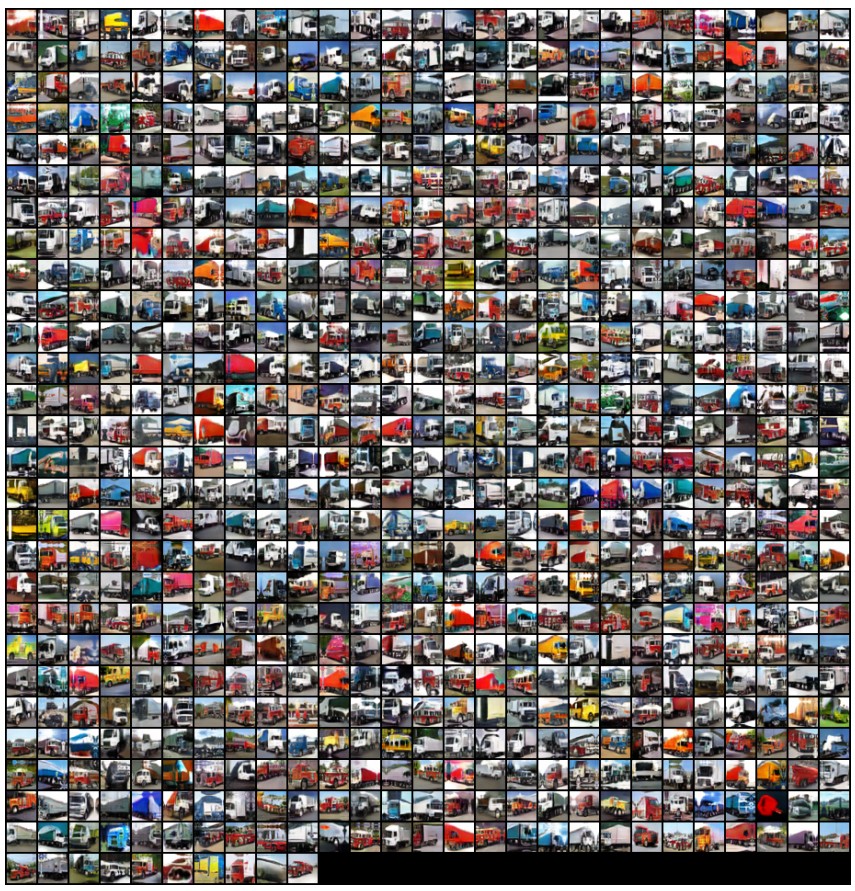

Figure 20: Class-conditional samples from CIFAR10

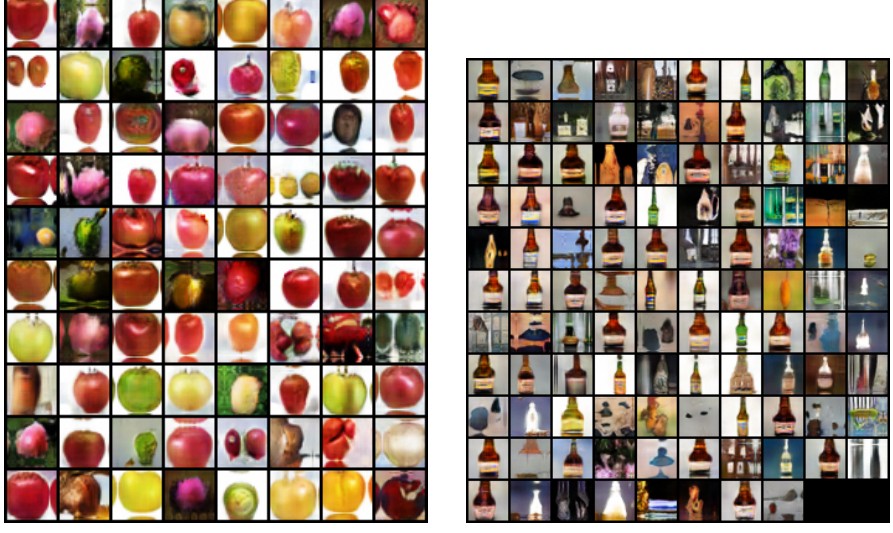

Figure 21: Class-conditional samples from CIFAR100

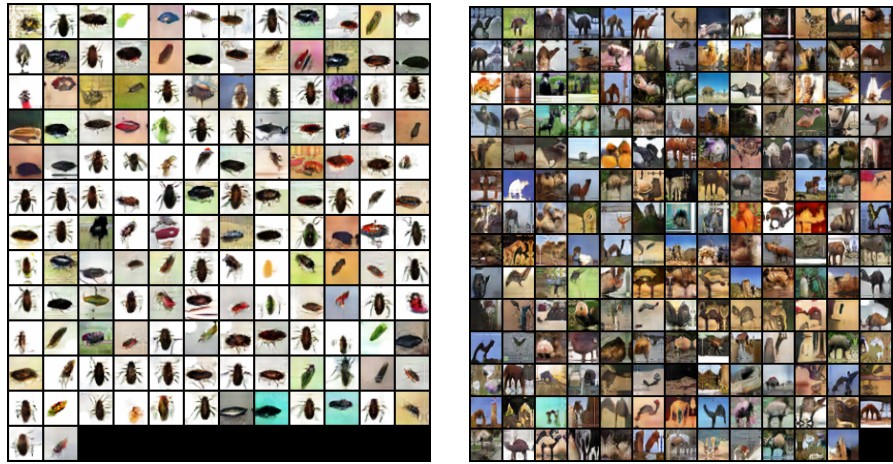

Figure 22: Class-conditional samples from CIFAR100

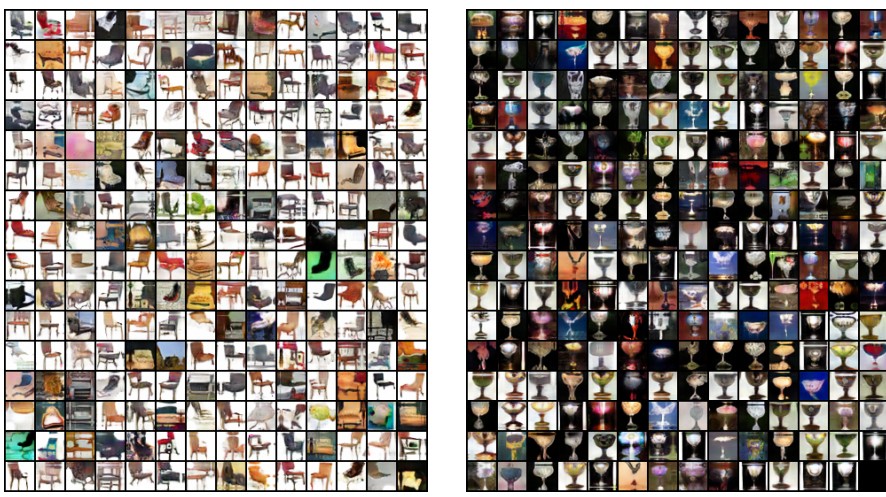

Figure 23: Class-conditional samples from CIFAR100

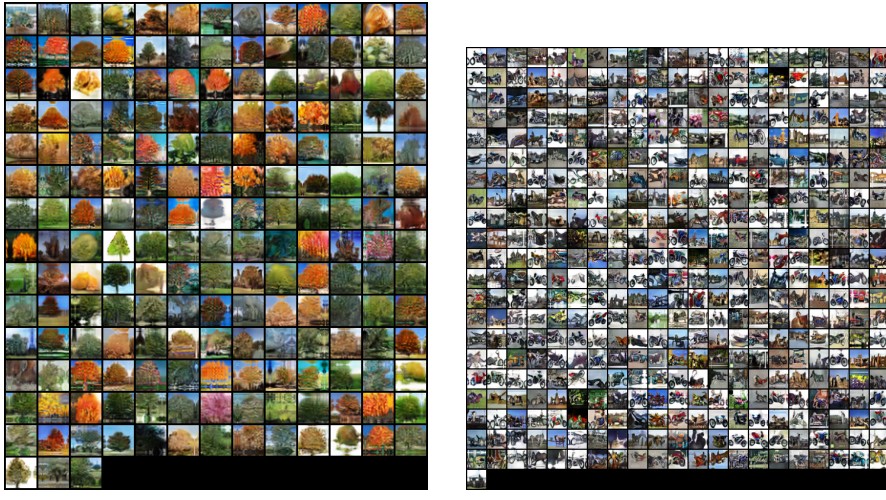

Figure 24: Class-conditional samples from CIFAR100

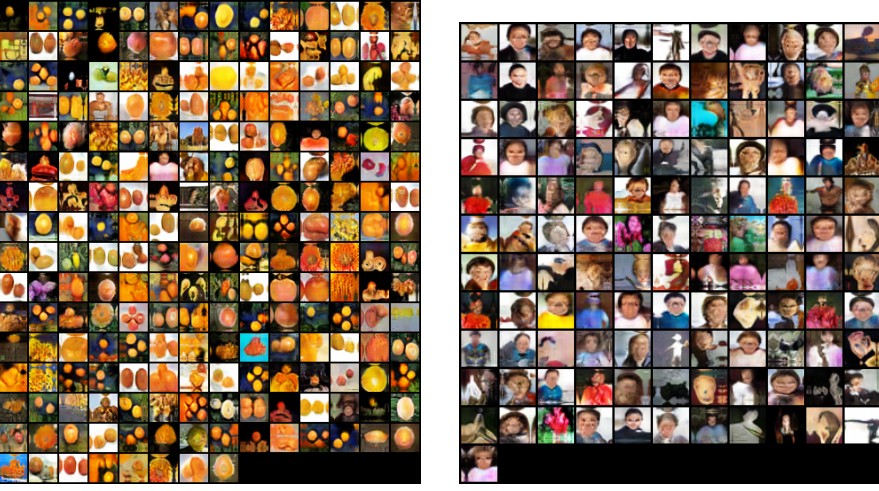

Figure 25: Class-conditional samples from CIFAR100

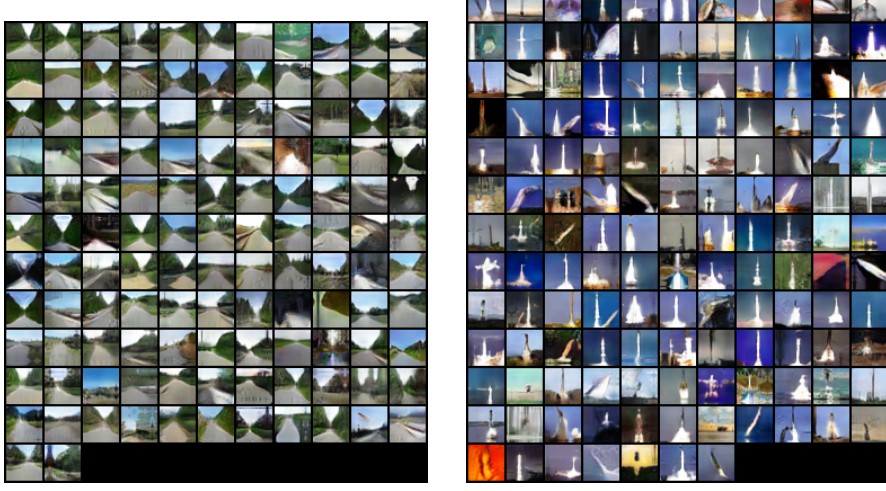

Figure 26: Class-conditional samples from CIFAR100

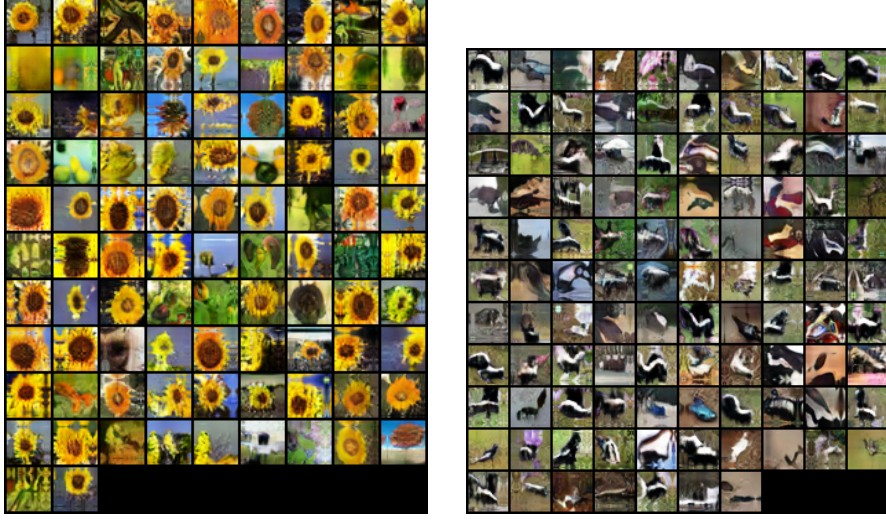

Figure 27: Class-conditional samples from CIFAR100

