# OpenReview forum: "No MCMC for me: Amortized sampling for fast and stable training of energy-based models"
_ICLR.cc/2021/Conference — ICLR 2021 Poster_

### Official Review · AnonReviewer2 · 2020-10-26
**An improved algorithm to train EBM-based models**

**Rating:** 4
**Confidence:** 4

**Review:**

This paper proposes an improved algorithm to train EBM-based models, called Variational Entropy Regularized Approximate maximum likelihood. The basic idea is to formulate the intractable partition function as an optimization problem with an additional entropy term. To estimate the gradient of the entropy term, the authors then propose a variational formulation for approximation.

The idea is interesting but not very exciting, and there consists of technical details that need to be carefully deal with. One of my biggest concerns is that since the algorithm relies on importance sampling to estimate the gradient of log marginal likelihood, the variance can be arbitrarily large. Specifically, the importance weight is the ration between a joint distribution and a proposed variational distribution \xi. One can see that a sample from \xi is a zero-mean Gaussian distribution, whereas q(z, x) should not. I suspect this would make the ratios to be vary unstable with high variance, which seems to contradict with the goal of the paper. In other words, I am not convinced why the importance sampling based method can work better than existing methods.

In addition, this importance sampling based method is introduced to avoid sampling the posterior distribution with HMC. I think this step can be considered as an efficient approximation to the  method, I am expecting HMC should perform at least as good as the proposed method but less efficient. The results in Table 2 seems to agree with this (although not completely). I wonder how is the running time comparison? Also, since HMC based method is a competitive method, why don't you consider this in other experiments such as those in Section 6?

Also, since the proposed method is claimed to outperform the recent SSM method. I think the same experiments as in the SSM paper should be conducted for comparison.

BTW, the generative images in Appendix are too small to be informative.

=========After rebuttal======
After the rebuttal, my main concern remains. Specifically, the paper defines a variational distribution q(z|x0 via a hierarchical construction: z_0 ~ N(0, 1), z ~ N(z_0, \eta I), which is essentially a zero-mean Gaussian. And I suspect the this is a bad variational distribution and it will induce high variance. The author said they didn't the hierarchical construction to define the variational distribution, because they fix z_0 after sampling. I don't think this is a formal way of defining a variational distribution. One reason is that even if they fix z_0, the proposal distribution will be a z_0-mean Gaussian, and the mean is randomly drawn from N(0, 1), which will not match the true posterior distribution (they only optimize the variance parameter). I think this should be make clear and investigated in more details. I will keep my initial score.

---

> ### Author Response · Authors · 2020-11-13
> **Thank you for the constructive feedback**
>
> We thank you for your time and your thoughtful comments on our work. We will address your concerns in order.
>
> Regarding the variance of the importance sampling estimator:
>
> These are important concerns you raise. You say in your review that $\xi$ is a sample from a zero-mean Gaussian distribution and this would be a bad proposal distribution for importance sampling from $q(z|x)$. We agree that if we were using such an uninformed proposal distribution for importance sampling, then the variance of the estimator would be very high. However, as described in the text directly following equation 8, this is not the proposal distribution we use. We construct an informed proposal distribution that closely approximates $q(z|x)$ and this informed proposal is a key contribution of our work. We elaborate more below:
>
> When generating a sample $x \sim q(x)$, we first sample $z \sim q(z)$ and then $x \sim q(x|z)$. The sample $z$ was thrown out in most previous approaches, but we do not do so. Knowing the $z$ which generated x allows us to build a much better proposal distribution. Namely, this $z$  by construction is a sample from the true posterior $q(z|x)$ and because of the smoothness of the generator network, the mass of $q(z|x)$ should concentrate near $z$. This same observation was made in [1] to motivate their use of only a few HMC samples to regularize the generator’s entropy. In our work, we let our proposal distribution be a Gaussian centered around $z$ and we further refine this proposal to maximize the ELBO which allows us to learn the dimension-wise scale of the proposal.
>
> Thus our proposal distribution is not a sample from a zero-mean Gaussian. It is a sample from a Gaussian centered around a true posterior sample whose dimension-wise scale is tuned to have minimal KL-divergence from the true posterior. To make this clearer, we have moved the definition of $\xi(z|z_0)$ out of the text into its own equation.
>
> In practice, we find importance sampling using this proposal performs very well (Table 2), even (slightly) out-performing HMC based entropy regularization. We attribute this improvement to a reduction in bias over the HMC entropy regularizer (Figure 3). We believe this gap in performance would likely close if more HMC steps were taken at training time but the number of samples used in our experiments (referring to figure 3, we used 2 burn-in steps) resulted in training which was 20x slower than our method (as mentioned in the paper).
>
> This massive increase in runtime is why we did not run the HMC baseline during our large-scale JEM experiments. The cost would have been prohibitive with the Resnet generator architectures we used.
>
> Despite all of this, we agree that the variance of the estimator is of just as much importance as the bias as analyzed in section 5.2. For that reason, we have added additional results to this section which analyze the variance of the estimator applied to the Factor Analysis model shown in Figure 3. We compare this variance with the HMC estimator. We find the stddev of our estimator is approximately 4x higher than the HMC estimator but the bias is approximately ⅕ the size of the HMC estimator (Figure 3). Empirically we find this increased variance to be alleviated by mini-batch averaging and the decreased bias results in better model performance. We also compute the effective sample size (ESS) of the importance sampling estimators. We find that the ESS of VERA is 1.31 for our CIFAR10 model and 1.29 for our MNIST model with the 20 importance samples we use in training, indicating importance sampling is giving informative gradient estimates. Using a $\mathcal{N}(0, 1)$ proposal gives an ESS of 1.0 (minimum possible value) for both MNIST and CIFAR10 models, indicating the uninformed proposal distribution gives uninformative gradient estimates (as expected).
>
> Regarding comparisons to experiments in SSM:
>
> We are confused by this question. Our experimental setup for section 5.1 exactly follows [2]. We stated this in Appendix B.3, but have also updated the paper to state this in the main body as well. The only difference is that we train our models for more epochs because of the reduced learning rates we use. We re-ran some of the SSM experiments as well to follow this number of training epochs but found them to obtain their best performance (in terms of likelihood) very early in training, so this change did not impact the results.
>
> The focus of this work was on large-scale EBM training so we do not feel that other experiments from the SSM paper are relevant to our method.
>
> Image size in the appendix:
>
> We have increased the size of these images.
>
> We hope this response and the changes we made have cleared up some of your concerns about our work. Thank you for your time.
>
> [1] Dieng, Adji B., et al. "Prescribed generative adversarial networks."
>
> [2] Song, Yang, et al. "Sliced score matching: A scalable approach to density and score estimation."

---

> > ### Comment · AnonReviewer2 · 2020-11-25
> > **Concern remains**
> >
> > Thanks for the detailed rebuttal. However, after reading the rebuttal, my concern on the proposal distribution being zero-mean gaussian remains.
> >
> > Specifically, you argue that the proposal distribution for q(z|x) is constructed as: z_0 ~ N(0, 1), z ~ N(z_0, \eta I). And you argue that the proposal distribution is not zero-mean gaussian. I don't get this, because after integrating out z_0, the proposal distribution is still a zero-mean gaussian. And you propose to optimize the parameter \eta, but that only affects the variance of the proposal distribution. As a result, this zero-mean Gaussian is likely to induce high variance of the proposed method. I think this is also verified by your additional experiments with HMC, where you show that your method has higher variance than HMC.
> >
> > Given this, I think this is a critical issue that does not convince me to change my rating.

---

> > > ### Author Response · Authors · 2020-11-25
> > > **Clarification**
> > >
> > > Reviewer,
> > >
> > > We believe that you misunderstand how our proposal distribution works. To generate a sample $x$ we first sample $z_0 \sim N(0, I)$ and then generate $x \sim N(g(z_0), \sigma  I)$. We look for a proposal distribution to approximate the true posterior of $p(z|x)$. Given that we know the true $z_0$ which generated $x$, we can use this to create a more informed proposal distribution. Since $\sigma$ is small, the space of possible $z$ values that could likely generate $x$ should concentrate near $z_0$. This indicates our knowledge of $z_0$ should be useful for building a posterior approximation.
> > >
> > > We only use this posterior approximation to regularize the generator's entropy. For this task, we only need to create a posterior approximation on samples generated from our model. This means that we have access to the exact $z_0$ which generated each sample we see. If we were trying to build a posterior approximation for data samples that we did not generate, then our approach would not work since we do not have access to the $z_0$ which generated $x$. Thankfully, we are only interested in building posterior approximation for samples that we generate.
> > >
> > > To reiterate, our posterior approximation is not a zero-mean Gaussian distribution. We re-use the exact $z_0$ which generated the $x$ that we see. We know this $z_0$ since we generated this $x$. We do not integrate out $z_0$, we generate it while sampling $x$ and then re-use the same value explicitly in our posterior approximation.
> > >
> > > We think this is the main point you are misunderstanding. If we were not to re-use the $z_0$ which generated $x$ and simply re-sampled it, then the proposal would be a zero-mean Gaussian as you say but that is not what we are doing.
> > >
> > > To address your other point about variance:
> > >
> > > Just because our estimator has larger variance than HMC does not mean it is of lower quality. As you can see in the plot above the one you mention, the estimator has notably lower bias. Variance can be dealt with (within reason) by increasing batch size but bias cannot.
> > >
> > > If it is possible, please respond before the discussion period is over. We will do our best to respond to you before the discussion period ends although it is late where we are.

---

> > > > ### Comment · AnonReviewer2 · 2020-11-25
> > > > **Not correct**
> > > >
> > > > I am afraid I cannot agree with the response. What you define for the proposal distribution is a hierarchical generative process. So z_0 is a random variable, saying that *we use the exact value* is not strictly correct. At the end, when talking about the proposal distribution, you have to integrate out all other random variables, which is the marginal distribution of z, and is a zero-mean gaussian. You cannot consider the conditional distribution as the proposal distribution.
> > > >
> > > > I also understand what you try to say in the response, and I think the current writing of the paper does not correspond to what you describe in the response. I think there are other ways to describe this but not sure how.

---

> > > > > ### Author Response · Authors · 2020-11-25
> > > > > **...**
> > > > >
> > > > > To our proposal distribution, $z_0$ is not a random variable. When we generate a sample $x$ there is exactly 1 $z_0$ which generates this $x$. There is a distribution of possible $z_0$ values which could create this $x$ but when we sample $x$ once, there is only one $z_0$. We are free to use this $z_0$ to condition our proposal distribution. This is described underneath eq. 8 in Section 4.2.
> > > > >
> > > > > The proposal distribution is not a hierarchical process.

---

> ### Author Response · Authors · 2020-11-14
> **Added new baselines**
>
> Thank you again for your thoughtful review. In response to the reviewer and public feedback, we have run a few more baselines which are now in an updated version of the paper.
>
> We ran MEG as a baseline for our SSL experiments. We find it performs on par with the supervised baseline. We believe this is due to the entropy regularizer being less effective than our own.
>
> We have also run CoopNets as a baseline in our NICE training section. It is a competitive approach, outperforming PCD and many SM variants, but it also has many of the same issues as PCD training. We found we needed to use considerably different hyper-parmeters than presented in the original work to stably train in this setting. Further, due to the MCMC sampling, CoopNets runs multiple times slower than VERA. We believe with more tuning or more MCMC steps, the results might be able to improve, but we leave exploring this further for future work.
>
> Thanks again for your time!

---

> ### Author Response · Authors · 2020-11-17
> **Response to feedback**
>
> Reviewer2,
>
> Again we thank you for your time and thoughtful review. We took your comments into consideration and we feel that we have greatly strengthened our paper. We hope that our response to your comments and the changes we have made in the paper have addressed your concerns. In particular, have we addressed your concerns regarding our importance sampling proposal and the variance of our estimator? We look forward to hearing from you.

---

### Official Review · AnonReviewer3 · 2020-10-27
**An extension of previous works, but demonstrated well**

**Rating:** 8
**Confidence:** 4

**Review:**

**Summary**: This paper presents a method for improving training of energy-based models. Rather than drawing samples using persistent contrastive divergence / MCMC, this approach parameterizes a separate model, which is trained to directly output samples. This effectively adds an additional KL divergence to the objective. The authors use a particular form of sampling model (a latent Gaussian model), borrowing a few tricks for getting entropy estimates out of the model. Results are demonstrated on a few qualitative setups, but most of the results are centered on improved JEM on sample quality, out-of-distribution detection, and semi-supervised learning. The main benefit of the approach seems to be speed and stability, however, the authors also claim that minimal tuning is needed.

**Strong Points**: Overall, the paper is fairly clear in its description of the background, method and experiments. Likewise, the tables, figures, and algorithm box provide clear demonstrations of key aspects of the approach. The descriptions in Eq. 6 are also useful for unfamiliar readers.

From my understanding, the technical aspects of the paper appear to be largely correct. Like previous papers, the authors use an objective that includes both the energy function and the approximate sampler. This includes the entropy of the sampler (from the sampler’s KL). The authors opt for a Gaussian latent variable model as the sampler. Evaluating the entropy thus entails an integral over the posterior of this model. The key technical contribution of this paper is to approximate this integral not with MCMC sampling, but instead use an importance-weighted variational approximation. This involves drawing a sample from the prior, then sampling points from a Gaussian around this point.

The experimental results in the paper are highly thorough. The authors demonstrate their method in settings where exact log-likelihood estimates are feasible (a flow-based model and probabilistic PCA). This allows the authors to demonstrate both improved samples and log-likelihood performance. In these cases, the authors compare with several relevant baselines. In the main experiments of the paper, the authors incorporate their approximate sampling scheme into JEM, a recent hybrid discriminative/generative model. Experimental results demonstrate that using the approximate sampler generally results in (slight) improvements over JEM, with the exception of semi-supervised classification, where the proposed method is substantially better. In all cases, experiments are performed using standard, benchmark datasets, and the authors compare with competitive, recent methods.

There are substantial details on the experiments in the appendix. This, combined with the experiments across multiple datasets and settings, suggests that these results are reproducible. However, given the fact that this methods requires training multiple models, exactly reproducing the results will likely require the corresponding code (which the authors did not provide).

While the method itself is not highly novel, the thorough experimental results somewhat compensate for this aspect. In particular, at least in the “toy” experiments, the authors compare with MEG, a relevant, related baseline. This demonstrates that, although the proposed method is a natural extension of previous works, the contribution is nevertheless useful.

**Weak Points**: One weak point of this paper is with regard to novelty. The main contribution of this paper is in estimating the integral of the sampling model’s entropy with an importance-weighted variational approximation. This is a fairly natural extension of previous works on learning sampling models, simply making the estimation procedure more efficient. On its own, this is not a highly substantial contribution. It’s also not clear that this is the best set of design choices for this problem. For instance, would the issues with the entropy gradient not be obviated with an exact log-likelihood model, such as a flow-based model / autoregressive model?

Some of the theoretical considerations of the setup appear to be skimmed over. The authors present the approximate sampling method with an equality in Eq. 4. However, in practice, the sampling model will not be able to reach the max, due to inherent limitations in the model class. Further, in Algorithm 1, the energy-based model and the sampling model are trained jointly, rather than training the sampling model inside of an inner loop (for the max operation). This is analogous to amortized inference in VAEs, where training the encoder and decoder jointly typically implies that the encoder cannot fully maximize the ELBO for the decoder. This phenomenon is referred to as “lagging inference networks,” (He et al., 2019) and the performance gap is the “amortization gap” (Cremer et al., 2018). Similarly, there will be a “sampling gap” here due to a “lagging sampling model.” While the method still seems to work well in practice, the fact that this is an approximation is almost entirely missing from the paper. Note that, to generate samples, the authors do indeed run additional MCMC steps (see appendix).

Finally, although the experiments are comprehensive, it’s not clear whether the improvements over JEM are highly substantial. In addition, in the case with the largest gap, semi-supervised learning, it’s unclear why the proposed method, VERA, should dramatically outperform JEM. Given that this is the most substantial result, it seems as though further investigation should be performed to analyze the reason for the boost in performance. Further, the experiments on JEM only compare the baseline models (which uses MCMC) with the proposed (amortized) method. A more complete set of experiments would also include other learned sampling methods, such as MEG.

**Accept / Reject**: Although the method is not highly novel, the experiments are fairly comprehensive in demonstrating the proposed method. Analyses are performed in multiple settings, across multiple datasets, comparing with the relevant baselines. At the very least, this paper demonstrates an efficient method that seems to outperform previous methods in simpler tasks and improve JEM on more difficult tasks. For these reasons, I would be in favor of accepting this paper. With a more detailed motivation/explanation of the design choices, discussion of the quality of the approximation, and some additional sampling model baselines on the JEM results, this paper could be further improved.

**Questions**:

Could you elaborate on the issue with the score function estimator in Eq. 6? Is this simply a matter of having a high-variance estimator?

In the importance weights, should $q_\phi (x)$ be included in the denominator?

In Algorithm 1, are you using a mini-batch of data examples but only one sample from the model. If so, why not use a batch of samples?

The MNIST samples don’t look great, even for VERA. Why is this the case?

Is using NICE in Section 5.1 representative of energy-based models? Flow-based models tend to have their own issues associated with stability.

Why is VERA better at semi-supervised learning as compared with JEM?

**Additional Feedback**:

Method:
I would not compare Eq. 5 with a GAN decoder, as GANs define an implicit density.
“or simple a diagonal Gaussian” —> “or simply a diagonal Gaussian”
“to the gradient of our model’s likelihood…”: should specify that this is the EBM.

EBM Training Experiments:
I would put zeros in front of all decimals to be consistent.
Not sure if it’s fair to conclude that entropy regularization is unnecessary for preventing mode collapse from this experiment. This seems like a limited setting.

Section A.2:
“Equation equation 7” —> “Equation 7”

Figure 7:
“Unconditional CIFAR10 Samples” —> “Unconditional MNIST Samples”

---

> ### Author Response · Authors · 2020-11-13
> **Thank you for the constructive feedback (1/2)**
>
> We thank you for your very comprehensive, thoughtful review, and for your kind words about our work.
>
> We will address your concerns in order:
>
> Regarding novelty:
>
> We have developed a novel entropy regularizer for latent-variable models which is considerably faster than prior approaches, does not require the training of auxiliary MI or score networks, and outperforms prior approaches for training large scale EBMs. Our proposed training procedure is several times faster than the state-of-the-art EBM training methods and alleviates the notorious stability issues found in these state-of-the-art approaches [1]. These stability improvements allow EBMs to be applied to domains where MCMC-based training fails (section 6.1) and to perform well at new applications. We believe our work provides a novel contribution to the field.
>
> Use of flows or AR models:
>
> Autoregressive models are not useful in this setting since sampling comes at an O(D) cost and we must sample at every training iteration. The cost of sampling would dominate training. Flows are not favorable since they are much less parameter efficient and produce lower quality samples than the latent-variable models we use. The CIFAR10 Glow [2] model has approximately 100x as many parameters as our Resnet generator and is much slower to sample from. Despite this, it achieves much lower sample quality (in terms of FID). For these reasons, we felt it valuable to address the difficulties that arise when using latent-variable generators -- which lead to this work.
>
> Regarding the approximations:
>
> In section 5.1 we explicitly state that our proposed optimization procedure maximizes an upper bound on likelihood. The objective is an upper bound exactly due to the “sampling gap” you bring up. The results in this section were specifically added to the paper to demonstrate that while we are indeed maximizing an upper bound, the bound is tight enough to train high-quality models at scale. To make this more clear we have added a discussion of this under equation 4 in the updated paper. We hope this clarifies your concern.
>
> Regarding your point about using MCMC to generate samples:
>
> The MCMC refinement procedure is there only to improve sample quality by making the generator samples match the distribution defined by the EBM more closely. This is not necessary and samples taken only from the generator are of high quality and competitive -- but of slightly lower quality. This can be seen in Appendix C.3.
>
> Regarding Improvements over JEM:
>
> The main intent of this paper was to present a new method for training EBMs which is faster and more stable than MCMC-based training. We chose to apply this to JEM as JEM is a new and effective application of EBMs. Training JEM as proposed in [1] is slow to train and unstable ([1], Sections 6, H.3). The main point of these experiments was to demonstrate that we can match the performance of JEM while training much faster and relieving the instability that is common with PCD training of JEM models. The fact that we improve upon the results is even better, but we feel our point would have been made even if the results were identical to the original work on JEM.
>
> PCD training of EBMs has made much progress but almost all research in this area has focused on image models. Little attention has been paid to alternate domains. As we believed our approach is more stable and easier to tune than PCD training we wanted to focus on training models outside of the image domain. This is why we focused on a diverse range of non-image datasets for our SSL experiments. Note that SSL outside of the image domain can be especially challenging because we can’t rely on strong prior knowledge of useful data augmentations. Our VERA-trained JEM models achieved strong results at SSL whereas our PCD training JEM models all performed poorly despite a considerable hyper-parameter search. We believe VERA’s improved performance in this setting is due to its stability and ease of tuning. We believe it is likely that a PCD trained model could achieve similar performance but we were not able to train one to convergence.
>
> We chose not to compare with MEG in this setting because of our results in section 5.1 which demonstrated that their entropy regularizer had little-to-no effect on MNIST sized data.
>
> ----continued below------

---

> > ### Author Response · Authors · 2020-11-13
> > **Thank you for the constructive feedback (2/2)**
> >
> >
> > Your questions:
> >
> > 1) The issue with Eq 6 is that this entropy estimator requires the score function of q(x). Unfortunately, q(x) is a latent-variable model and the score function cannot be evaluated easily. The issue with the score function estimator in Eq 7 is that it requires posterior samples which also cannot be easily computed and we must resort to MCMC or VI techniques to approximate the expectation in Eq 7.
> >
> > 2) We define how the importance weights are computed in the text after Eq 8. There, you can see q(x) is not used in computing the importance weights. We include q(x) in the derivation since this is how the posterior can be defined using Bayes rule.
> >
> > 3) We intended x_g to refer to a mini-batch of samples from the generator. We have updated Algorithm 1 to make this more clear.
> >
> > 4) These samples do not look great because they are exact samples from a NICE model trained in a way that ignores its normalizing constant. This experiment is meant to demonstrate the quality of various training methods that do not require a normalized model. If we did not care about the likelihood we could generate higher quality samples, but this was not the intent of this experiment. The main goal here was to demonstrate that approximate (generator) samples from VERA matched exact samples much more closely than approximate samples from other training procedures such as PCD and MEG. You can see MNIST samples from our VERA SSL model in Appendix C.2 which are of much higher quality.
> >
> > 5) We agree that flow-based models have their own issues, but the point of this experiment was simply to demonstrate that models trained with VERA achieve higher likelihood than alternative training approaches that do not require a normalized model. Results like this are commonly presented when proposing a new method for training unnormalized models. We typically train a model, whose likelihood can be computed, with our approach and then evaluate the training methods using the true model likelihood. Previously linear ICA was the preferred model [3, 4, 5, 6], but normalizing flows present a more interesting class of models to explore since, unlike linear ICA, they are capable of accurately modeling complex, high dimensional data such as images. [7] evaluates their training method using these models and we felt it useful to train these models with VERA since this is the most challenging model and dataset we were aware of where these results have been benchmarked.
> >
> > 6) We have elaborated on our SSL results above.
> >
> > Your additional notes:
> >
> > “I would not compare Eq. 5 with a GAN decoder, as GANs define an implicit density.”
> >
> > We have removed this sentence.
> >
> > “Not sure if it’s fair to conclude that entropy regularization is unnecessary for preventing mode collapse from this experiment. This seems like a limited setting.”
> >
> > We’ve qualified this claim with “in this setting”.
> >
> > Thank you for pointing out these minor inaccuracies. We agree with all of them and have updated the paper accordingly.
> >
> > With regards to your comment on reproducibility, here is an anonymized Google Drive link to our code: https://drive.google.com/drive/folders/1y1lBAtt5fzp4IgZ4URuYmUgRHCIBGag7?usp=sharing
> >
> > [1] Grathwohl, Will, et al. "Your classifier is secretly an energy based model and you should treat it like one."
> >
> > [2] Kingma, Durk P., and Prafulla Dhariwal. "Glow: Generative flow with invertible 1x1 convolutions."
> >
> > [3] Hyvärinen, Aapo. "Estimation of non-normalized statistical models by score matching."
> >
> > [4] Gutmann, Michael, and Aapo Hyvärinen. "Noise-contrastive estimation: A new estimation principle for unnormalized statistical models."
> >
> > [5] Ceylan, Ciwan, and Michael U. Gutmann. "Conditional noise-contrastive estimation of unnormalised models."
> >
> > [6] Grathwohl, Will, et al. "Learning the Stein Discrepancy for Training and Evaluating Energy-Based Models without Sampling"
> >
> > [7] Song, Yang, et al. "Sliced score matching: A scalable approach to density and score estimation."

---

> ### Author Response · Authors · 2020-11-14
> **Added new baselines**
>
> Thank you again for your thoughtful review. In response to the reviewer and public feedback, we have run a few more baselines which are now in an updated version of the paper.
>
> We ran MEG as a baseline for our SSL experiments. We find it performs on par with the supervised baseline. We believe this is due to the entropy regularizer being less effective than our own.
>
> We have also run CoopNets as a baseline in our NICE training section. It is a competitive approach, outperforming PCD and many SM variants, but it also has many of the same issues as PCD training. We found we needed to use considerably different hyper-parmeters than presented in the original work to stably train in this setting. Further, due to the MCMC sampling, CoopNets runs multiple times slower than VERA. We believe with more tuning or more MCMC steps, the results might be able to improve, but we leave exploring this further for future work.
>
> Thanks again for your time!

---

> ### Author Response · Authors · 2020-11-17
> **Response to feedback**
>
> Reviewer3,
>
> Again we thank you for your time and thoughtful review. We took your comments into consideration and we feel that we have greatly strengthened our paper. We hope that our response to your comments and the changes we have made in the paper have addressed your concerns. We look forward to hearing from you.

---

### Official Review · AnonReviewer1 · 2020-10-29
**An overall nice idea, but can be stronger with more comparison/dicussion with previous work.**

**Rating:** 7
**Confidence:** 4

**Review:**

This paper proposes a new method on training energy-based models with maximum likelihood. Instead of using MCMC approaches to sample from the EBM, authors follow previous work on training neural generators for faster sample generation. In particular, authors consider a special generator where the output is convolved with Gaussian noise. The score function of this generator can be estimated with self-normalized importance sampling, which is then used to estimate the entropy term through the reparameterization trick. Authors demonstrate that their new method is able to train EBMs efficiently, and improves the stability and performance of JEMs compared to MCMC-based training approaches.

#### Pros
* The method is more computationally efficient compared to MCMC-based approaches. As the title suggests, no iterative MCMC approaches are needed. Though the technique is inherently similar to Dieng et al. (2019), authors replaces the HMC sub-routine with a carefully designed variational approximation.

* Experiments on JEMs are particularly interesting. Authors demonstrate that their method can train JEMs in a stable way and outperforms baselines on classification accuracy, sample quality, out-of-distribution detection and semi-supervised learning.

* Writing is clear and easy to follow.

#### Cons
* From my perspective, the biggest disadvantage is related to various additional hyper-parameters. In VERA training, $\gamma$ controls the gradient penalty and $\lambda$ controls the contribution of the estimated entropy gradient. Both requires considerable tuning for optimal performance. However, typical MCMC-based approaches do not require gradient penalty, and tuning $\lambda$ is unsatisfying — shouldn't $\lambda \equiv 1$ for real maximum likelihood training?

* The idea of training a neural generator with the dual form of the likelihood objective has been explored before. Authors have compared with MEG, which uses a similar objective. I think adversarial dynamics embedding is also a necessary baseline, since it uses the same objective and has a special design of the neural generator to make entropy computing tractable. It would be better if authors will include it in both NICE and JEM experiments.

* Authors cited (Song & Ermon, 2019a) when pointing out the difficulty of MI estimation. The reference below should also be included as it actually appeared earlier in 2018.

McAllester, David, and Karl Stratos. "Formal limitations on the measurement of mutual information." International Conference on Artificial Intelligence and Statistics. 2020.

* Authors did not include methods such as MEG in the JEM experiments. Any reason?

----------------
Post-rebuttal

I appreciate the authors' response and additional comparison against previous work. I do think that proper comparison with previous work is important, as it allows us to know better when and where the proposed approach is beneficial.

---

> ### Author Response · Authors · 2020-11-13
> **Thank you for the constructive feedback**
>
> We thank you for your time and your kind words about our work. We will address your concerns in order.
>
> Regarding hyper-parameters:
>
> While we agree our approach introduces a new set of hyper-parameters to be tuned, we disagree that our approach “adds” hyper-parameters. In fact, our approach has fewer hyper-parameters than MCMC-based EBM training methods and these hyper-parameters can be set using common-sense principles.
>
> As you point out, the main hyper-parameters to tune are the strength of the entropy regularizer $\lambda$ and the strength of the gradient regularizer $\gamma$.
>
> We compare this with the PCD training used in the original JEM work [1]. In PCD training with SGLD you must specify each of: the SGLD step-size, number of MCMC steps, SGLD noise variance, buffer size, and reinitialization frequency. So PCD has 3 more hyper-parameters than VERA.
>
> Further, we add that the gradient regularizer strength was chosen early in our experiments and not searched over. We found this choice worked well across multiple data domains and model sizes, so we recommend setting $\gamma = .1$ and ignoring this when searching hyper-parameters.
>
> Further still, a simple analysis can verify that the strength of the entropy regularizer $\lambda$ has the effect of tempering the energy function. This is analogous to decoupling the step-size in SGLD from the noise variance as is common in EBM training [1, 2, 3, 4]. This has the effect of increasing the impact of the gradient of the energy function in generator training (or in sampling for PCD) which makes learning more efficient at the cost of sample quality. As in PCD, this can be tuned to be as large as possible while keeping training stable. We have added a section (B.1.1) to the appendix to explain this connection.
>
> We also find that training with VERA allows us to remove the Gaussian noise typically added to the input data to stabilize EBM training. The scale of this noise is another important hyper-parameter to set [1, 2] and it can be removed completely when training with VERA.
>
> So to summarize, VERA has fewer hyper-parameters than PCD training, one parameter does not need to be searched over, the other parameter can be tuned in a common-sense fashion, and we can completely remove a pre-preprocessing step (which has its own hyper-parameters) that was previously required for stable EBM training.
>
> Thus we feel that the number of hyper-parameters of VERA is actually a strength of our method instead of a weakness.
>
>
> Regarding MEG in the JEM experiments:
>
> As you can see in section 5.1 the results of MEG are near identical to training with no entropy regularization at all. Further, our experiments on mode-capturing demonstrate that entropy regularization is not necessary to achieve strong performance at this task on MNIST-sized data. Thus, we feel that the MEG entropy regularizer is not responsible for the published performance at this task (and no baseline without their regularizer was reported in the original work on MEG). This, combined with the results from 5.1 led us to believe this approach to entropy regularization would not be competitive in high dimensions so it was left out of our comparisons.
>
> Regarding Adversarial Dynamics Embedding in NICE and JEM experiments:
>
> We did not compare against this method as it was complex, not fully described in their paper, and the code provided was not easy to use.
>
> Regarding the citation for MI estimation:
>
> We were not aware of this work. It is quite insightful, and we have added a reference to the updated paper.
>
> [1] Grathwohl, Will, et al. "Your classifier is secretly an energy based model and you should treat it like one."
>
> [2] Nijkamp, Erik, et al. "Learning non-convergent non-persistent short-run MCMC toward energy-based model."
>
> [3] Nijkamp, Erik, et al. "On the anatomy of mcmc-based maximum likelihood learning of energy-based models."
>
> [4] Du, Yilun, and Igor Mordatch. "Implicit generation and generalization in energy-based models."

---

> ### Author Response · Authors · 2020-11-14
> **Added new baselines**
>
> Thank you again for your thoughtful review. In response to the reviewer and public feedback, we have run a few more baselines which are now in an updated version of the paper.
>
> We ran MEG as a baseline for our SSL experiments. We find it performs on par with the supervised baseline. We believe this is due to the entropy regularizer being less effective than our own.
>
> We have also run CoopNets as a baseline in our NICE training section. It is a competitive approach, outperforming PCD and many SM variants, but it also has many of the same issues as PCD training. We found we needed to use considerably different hyper-parmeters than presented in the original work to stably train in this setting. Further, due to the MCMC sampling, CoopNets runs multiple times slower than VERA. We believe with more tuning or more MCMC steps, the results might be able to improve, but we leave exploring this further for future work.
>
> Thanks again for your time!

---

> ### Author Response · Authors · 2020-11-17
> **Response to feedback**
>
> Reviewer1,
>
> Again we thank you for your time, thoughtful review, and defense of our work. We took your comments into consideration and we feel that we have greatly strengthened our paper. We hope that our response to your comments and the changes we have made in the paper have addressed your concerns. We look forward to hearing from you.

---

### Public Comment · ~Feng_Shi1 · 2020-11-14
**The current idea has been published as the CoopNets algorithm**

Dear authors, reviewers, and AC,

The paper claims that it proposes “using a generator as amortized sampling for fast training of Energy-based models” as its main contribution. I think this is exactly what the paper “cooperative network (CoopNet)”[1] does.

Specifically, the cooperative network (CoopNet) in [1] jointly trains two components:  a generator G (the student net), which learns to be a fast sampler for amortizing MCMC, and a descriptor net D (the teacher net), which is an energy-based model to distill the knowledge to the generator.  The generator helps the EBM for MCMC, and the EBM teaches the generator. The cooperative training simultaneously alternates the MLE training of both EBM and the generator. Both models are parameterized by deep neural networks. Such a training scheme is well known as “cooperative learning”.

It is also in a textbook [2] used in UCLA for graduate course stat 202C:

http://www.stat.ucla.edu/~sczhu/Courses/UCLA/Stat_202C/Stat_202C.html

I can share some related pages about the cooperative learning in that book here since the book is not free:  https://bit.ly/mc_book_sample

Other papers using Cooperative Nets include [3], which is learning a conditional generator for amortizing MCMC of a conditional energy-based model for conditional learning.

I believe there are still other related research articles using such a framework in other domains. Therefore, I suspect the novelty of the current paper under review.

My concerns and questions are:

To authors:

(1)    I don’t know why the authors didn’t discuss, cite, or even mention the existing framework. I found that most of the current ConvNet-EBM-related papers only cited prior works from 2019, and totally ignored those papers that first proposed the original model and learning algorithm. This is just like, we will not cite Goodfellow’s 2014 GAN paper anymore, but only cite BigGAN or progressive GAN when we refer to adversarial training algorithms.

(2)    If the authors were unaware of the work I mentioned above before, then given the fact that your current framework belongs to the existing cooperative training framework or its variant. I suggest the authors call your method “xxx_CoopNets”. Re-naming an existing framework is not encouraged. This is just like, we re-name GAN with other names.


To reviewers and AC:

(1)    Given the fact that the idea in the current paper bears a strong similarity with the CoopNets paper [1], I think you might need to re-evaluate the novelty of the paper.

(2)    If the paper is finally accepted for some reason, do you think the paper should use the name that has been used in the original paper [1] or re-name everything as a new one? This is a very general question that frequently happens to any of us. Because this is about how we should grant credits to the prior works.

References:
[1] Xie et al. Cooperative Training of Descriptor and Generator Networks. (TPAMI) 2018.
https://arxiv.org/pdf/1609.09408.pdf

[2] Adrian Barbu, Song-Chun Zhu. Monte Carlo Methods. Springer 2020.
https://bit.ly/mc_book_sample

[3] Xie et al. Cooperative Training of Fast Thinking Initializer and Slow Thinking Solver for Multi-Modal Conditional Learning. https://arxiv.org/pdf/1902.02812.pdf

---

> ### Comment · AnonReviewer1 · 2020-11-14
> **In defense of the authors**
>
> Thanks for bringing additional related work to our attention. I agree with Feng Shi and Yang Lu that previous work on cooperative training is relevant and I think it probably should be included as a baseline for experimental comparison.
>
> There are several points, however, that I do not fully agree with.
>
> First, it is pretty clear that this paper did NOT claim to be the **first one** that "uses a generator as amortized sampling for fast training of EBMs". Authors acknowledged that they use the same objective as in Kumar et al. (2019), Dai et al. (2019), which both train a generator to amortize sampling for EBM training. The major contribution of this paper is another method for optimizing the dual objective, not (and the authors never claimed) the idea of amortized neural generators for EBM training. On that note, authors should probably also cite Kim & Bengio (2016) and Zhai et al. (2016), which are even more relevant than CoopNet.
>
> Second, CoopNet is not the first paper that proposes the idea of amortized sampling for EBM training. As far as I know, Kim & Bengio (2016) seems to be the earlier one that tries to accelerate EBM training with a neural generator. For this reason, I don't think it makes sense to name every related method "xxxx_CoopNet". Even if CoopNet was the earliest, such a naming requirement seems very weird to me.
>
> Finally, the method in this work is sufficiently different from CoopNet, and has its own novelty. CoopNet essentially uses two separate objectives for training the EBM and the generator, while this work (and previous ones on the dual form of likelihood) has a unified objective. In addition, CoopNet still needs multiple steps of Langevin MCMC despite having a generator, which is also different from this paper.

---

> > ### Author Response · Authors · 2020-11-14
> > **Thank you (again) for defending our work.**
> >
> > We thank you very much for taking the time to understand the key differences between our proposed method and Coopnets. We have already updated our paper to add a citation to Coopnets and discuss the similarities and differences.

---

> ### Author Response · Authors · 2020-11-14
> **Related work**
>
> Feng,
>
> We were first made aware of CoopNets after Yang's comment yesterday. We appreciate you as well bringing it to our attention. As we said in our response to Yang, we completely agree that CoopNets are related to our work and to the many other works which have been published in the past few years (some others even submitted to this conference) which train EBMs and alongside generators. We sincerely apologize for not originally citing and comparing with CoopNets. We have remedied this in the updated paper.
>
> Beyond this, we disagree with the tone and content of your comments. We made no claims that our work is the first to train EBMs along with a generator. You also seem to overlook the many things that make our work distinct from CoopNets. First, we use a different training objective. Second, the focus of our work is to develop an approach for training large-scale EBMs without any MCMC sampling (hence our paper's title). CoopNets require MCMC sampling to train both the EBM and the generator. Removing MCMC provides the challenge of generator entropy estimation which we tackle, leading to the main contribution of our work -- a fast, efficient, effective entropy regularizer. This contribution is completely orthogonal to CoopNets.
>
> Putting this together, we develop an MCMC-free method for training EBMs. We demonstrate the utility of this training approach in a number of distinct applications, model classes, and evaluation metrics.
>
> For these reasons, we disagree with you about the novelty of our work.
>
> Further, we disagree that any method which includes a generator and an EBM should be named a "CoopNet." Our training objective, intended application, and many other factors differ from CoopNets, so we do not feel this warrants a name change.  As reviewer1 pointed out, CoopNets are not the first published work to do this. Further, the Inclusive-NRF of [1] is more related to CoopNets than our work and yet is not referred to as a CoopNet. Do you also propose Maximum Entropy Generators [2], Adversarial Exponential Family Dynamics Embedding [3], FLow-Contrastive Estimation [4],  and Auxiliary-variable Local Exploration [5] also be renamed as CoopNets?
>
>
> We appreciate you and Yang bringing this work to our attention. We believe the added discussion and new experiments greatly strengthen our work. Thank you very much for that.
>
>
>
>
> [1] Song, Yunfu, and Zhijian Ou. "Learning neural random fields with inclusive auxiliary generators."
>
> [2] Kumar, Rithesh, et al. "Maximum entropy generators for energy-based models."
>
> [3] Dai, Bo, et al. "Exponential family estimation via adversarial dynamics embedding."
>
> [4] Gao, Ruiqi, et al. "Flow contrastive estimation of energy-based models."
>
> [5] Dai, Hanjun, et al. "Learning Discrete Energy-based Models via Auxiliary-variable Local Exploration"

---

> ### Comment · ~Philip_Bachman1 · 2021-03-11
> **CoopNets was a dupe too**
>
> FYI, we had a paper at ICLR 2017 which also optimized this objective. It was called "Calibrating Energy-Based Generative Adversarial Networks". https://arxiv.org/abs/1702.01691

---

> > ### Comment · ~Jianwen_Xie1 · 2021-07-20
> > **CoopNets is NOT a dupe**
> >
> > FYI, the CoopNets was posted in arXiv on 29 Sep 2016, https://arxiv.org/pdf/1609.09408v1.pdf.  The paper of "Calibrating Energy-Based Generative Adversarial Networks" was posted in arXiv on 6 Feb 2017.  The CoopNets is earlier.

---

### Public Comment · ~Zhisheng_Xiao1 · 2020-11-14
**Training on unconditional EBM rather than JEM?**

Hi authors! I like your new idea on training EBMs. One question: your experiments on real image dataset is training a JEM, while the result is impressive, I may wonder that training JEM is easier than training an completely unconditional EBM, as the classification part can provide strong signal to learn the parameters. Is it possible to train unconditional EBMs (say, on images with no label like LSUN/CelebA, or at least just pretending we don't have labels for cifar-10) using your method? That might be a more challenging task than JEM. I personally think that to show the effectiveness of a new deep EBM training method, it would be better to disentangle all other factors and focus on the most basic generative modeling task. EBM training paper like [1,2,3] all performs at least some unconditional generation experiments.

[1]Implicit Generation and Modeling with Energy Based Models
[2] Learning Energy-Based Models in High-Dimensional Spaces with Multi-scale Denoising Score Matching https://arxiv.org/abs/1910.07762
[3]Training Deep Energy-Based Models with f-Divergence Minimization https://arxiv.org/abs/1910.07762

---

> ### Author Response · Authors · 2020-11-14
> **Thanks! Interesting ideas!**
>
> Zhisheng,
>
> Thanks for your interest in our work. These are interesting points you bring up. This was in a sense the goal of our NICE experiments. There we examine the ability of our approach to train high-quality, unconditional energy functions. We chose to use flows since we can easily quantify the quality of the models we learn using likelihood. We could have trained unconditional image models as well but the evaluation and comparison is a bit trickier. Most image models like this are evaluated using qualitative measures of sample quality (such as FID, IS). It's well known that sample quality does not necessarily line up well with likelihood. Further, if sample quality is the target measure, we believe other classes of models such as GANs and diffusion models are more promising. We are more interested in EBMs for their application to downstream tasks, hence the evaluation on JEM and SSL. Having said all this, I agree that it would be interesting to see these results and I may investigate that sometime soon.
>
> We hope this makes things more clear. Thanks again for you comment.

---

### Public Comment · ~ZENGYI_LI1 · 2020-11-16
**Yet another suggested reference/comparison**

I know this is frustrating, but I happen to know yet another paper [Abbasnejad et al 2019], which I'm pretty sure uses the same objective as your paper. Their approach of optimizing the entropy term is indeed different, but a comparison and some discussion is probably needed here.

Thanks


Reference:
Abbasnejad, M. Ehsan, et al. "A generative adversarial density estimator." Proceedings of the IEEE Conference on Computer Vision and Pattern Recognition. 2019.

---

> ### Author Response · Authors · 2020-11-16
> **more related work**
>
> Zengyi,
>
> Thank you for alerting us to this work. It is quite interesting and indeed related. We have updated our paper to include a citation and a discussion comparing this work to ours.

---

### Comment · ~Zhijian_Ou1 · 2021-03-14
**Some concerns**

Nice to read this paper. The paper addresses the important problem of improving EBM training and does a good job in demonstrating the usefulness of EBM and the new training method for a number of interesting applications. I have some concerns.

1. The claim on "no MCMC for me" is prone to misleading. The authors compare different EBM training approaches in the Introduction and Table 1. But there are some inaccurate points. First, MCMC methods can also use auxiliary models, such as in CoopNet, Inclusive-NRF, and so on. Second, it may be too simple to category MCMC methods as slow training. In fact, the time complexity of some MCMC methods USED IN TRAINING is not at all slow. It should be stressed that based on the stochastic approximation (SA) framework of model training [a,b], we do not need to run the Markov chain for sufficiently long time to converge within one SA iteration (theoretically one step Markov move would suffice). This is unlike in applications of MCMC solely FOR INFERENCE. This understanding is often overlooked when commenting on MCMC methods for model learning. Further, if a concern of this work is to improve training time complexity, then the readers expect this paper to provide some comparisons. In fact, Inclusive-NRF [c] uses one step Stochastic Gradient Langevin Dynamics (SGLD), much faster than IGEBM [d] and NCSN [e], and obtains superior image generation results (See below).

2. Note that on CIFAR-10, unsupervised Inclusive-NRF obtains IS 7.54±0.10, FID 27.9±0.53 [c]. In contrast, VERA uses the same CNN generator, more complexed Wide ResNet for energy function, and, remarkably, trains JEM which uses labels. Training JEM with labels is easier than training EBM. In fact, the JEM model is the supervised Inclusive-NRF in [c], and EBM is the unsupervised Inclusive-NRF. As shown in [c], supervised model significantly outperform unsupervised model. Considering these, the result of VERA in Table 4 is not so impressive.

[a] Andrieu, C., Moulines, É., and Priouret, P. (2005). Stability of stochastic approximation under verifiable conditions. SIAM Journal on control and optimization, 44(1):283–312.

[b] Song, Q., Wu, M., and Liang, F. (2014). Weak convergence rates of population versus single-chain stochastic approximation MCMC algorithms. Advances in Applied Probability, 46(4):1059–1083.

[c] Yunfu Song, Zhijian Ou. Learning neural random fields with inclusive auxiliary generators. arXiv:1806.0027, 2018.

[d] Yilun Du and Igor Mordatch. Implicit generation and generalization in energy-based models. arXiv:1903.08689, 2019.

[e] Yang Song, Stefano Ermon. Improved techniques for training score-based generative models. arXiv:2006.09011, 2020.

Thanks for the nice work.

---

> ### Author Response · Authors · 2021-03-14
> **some responses**
>
> Zhijian,
>
> The point of this work was not to completely demonize the use of MCMC for training energy-based models. MCMC-based training *can* be slow and hard to tune, making it hard to generalize to many domains (like our tabular data experiments). Your inclusive-NRF is not slow to train and gets very impressive results, but it does require the use of 2 different MCMC samplers which must be tuned. We must choose the stepsizes, the number of steps, type of sampler, and so on. These must be tuned properly or else the model will not train.
>
> We don't claim that this makes our approach superior to yours.
>
> We were interested in providing an alternative method for training EBMs that does not require the tuning of MCMC samplers which has been the most difficult part of EBM training in my personal experience.
>
> I hope this clarifies things.

---

### Decision · Program_Chairs · 2021-01-07
**Final Decision**

**Decision:**

Accept (Poster)

**Comment:**

The authors proposed to train an energy based model with a hierachical
variational approximations. The entropy can be tricky in hierarchical
variational approximations.  The authors suggest using the auxillary
samples to guide an importance samples to compute the gradient of the
entropy. They evaluate their approach on a slew of models. The idea is
straightfoward and could potentially be applied to other hierarchical
variational models out side of the energy-based model setting.  The
authors were responsive and clarified many agressive questions. I'd
ask the authors to clean up two things

- Equation 8 would be easier to follow if it kept the expectation from
  equation 6 thereby making z_0 feel like it materialize out of thin
  air


- A more detailed discusion of when the proposal is good and what could
  be missed out	when relying on the generating z to center the proposal